# Harmonizing Geometry and Uncertainty: Diffusion with Hyperspheres

**Muskan Dosi** [1]   **Chiranjeev Chiranjeev** [1]   **Kartik Thakral** [1]   **Mayank Vatsa** [1]   **Richa Singh** [1]

## Abstract

*Do contemporary diffusion models preserve the class geometry of hyperspherical data?* Standard diffusion models rely on isotropic Gaussian noise in the forward process, inherently favoring Euclidean spaces. However, many real-world problems involve non-Euclidean distributions, such as hyperspherical manifolds, where class-specific patterns are governed by angular geometry within hypercones. When modeled in Euclidean space, these angular subtleties are lost, leading to suboptimal generative performance. To address this limitation, we introduce **HyperSphereDiff** to align hyperspherical structures with directional noise, preserving class geometry and effectively capturing angular uncertainty. We demonstrate both theoretically and empirically that this approach aligns the generative process with the intrinsic geometry of hyperspherical data, resulting in more accurate and geometry-aware generative models. We evaluate our framework on four object datasets and two face datasets, showing that incorporating angular uncertainty better preserves the underlying hyperspherical manifold. Resources are available at: Link.

## 1. Introduction

Diffusion models have revolutionized generative modeling, achieving remarkable success in diverse modalities, including generation of images (Ho et al., 2020; Dhariwal & Nichol, 2021), audio (Kong et al., 2021), 3D scenes and 3D structures (Bautista et al., 2022; Shue et al., 2023). These models operate by progressively adding Gaussian noise to the data in a forward process, followed by a reverse process that learns to recover the original data from the corrupted versions (Sohl-Dickstein et al., 2015). The use of Gaussian

noise, coupled with its isotropic nature, simplifies both theoretical formulations and practical implementations, making it a default choice for most diffusion frameworks (Kingma et al., 2021). However, this assumption inherently biases the data to Euclidean spaces (Song et al., 2021; Dhariwal & Nichol, 2021), limiting the model's ability to account for intrinsic geometric structures in non-Euclidean domains (Lui, 2012; Scott et al., 2021; De Bortoli et al., 2022), such as hyperspherical or manifold-constrained data (Bronstein et al., 2017; Rezende et al., 2020). In such settings, Gaussian-based diffusion may fail to fully capture the directional relationships and nuanced variations inherent to the data, as shown in Figure 1(a) (top row) where the forward process of adding Gaussian noise leads to distortion in the angular geometry of the classes during sampling. It motivates a need for rethinking the noise distribution in the diffusion processes.

Beyond the limitations of geometry, Gaussian noise also overlooks the flexibility required to model varying uncertainty levels across different samples. Ambiguous or noisy data points appear across different timesteps in Figure 1(a) (top row). These points require a representation where uncertainty is explicitly modeled. However, the isotropic assumption of Gaussian noise treats all directions equally, failing to capture this uncertainty effectively. Researchers have explored alternative noise distributions better suited to specific data geometries and uncertainty modeling (Xu et al., 2023). One promising direction is the von Mises-Fisher (vMF) distribution (Mardia & Jupp, 2000), which is naturally defined on hyperspheres and parameterized by a concentration parameter that controls directional uncertainty (Hasnat et al., 2017). For effective handling of uncertainty using vMF noise in diffusion models, we aim to align the generative process with the underlying geometry of the data, thus enabling more accurate modeling of directional relationships. This approach builds on recent efforts to integrate geometric priors into generative models (Falorsi et al., 2018), expanding the scope of diffusion techniques to hyperspherical data.

### 1.1. Research Contributions

We propose **HyperSphereDiff** (Diffusion with Hyper-Spheres), leveraging the von Mises-Fisher (vMF) distribution to preserve hyperspherical class geometry by introduc-

---

[1]Department of Computer Science and Engineering, Indian Institute of Technology Jodhpur, India. Correspondence to: Muskan Dosi <dosi.1@iitj.ac.in>.

*Proceedings of the 42ⁿᵈ International Conference on Machine Learning*, Vancouver, Canada. PMLR 267, 2025. Copyright 2025 by the author(s).

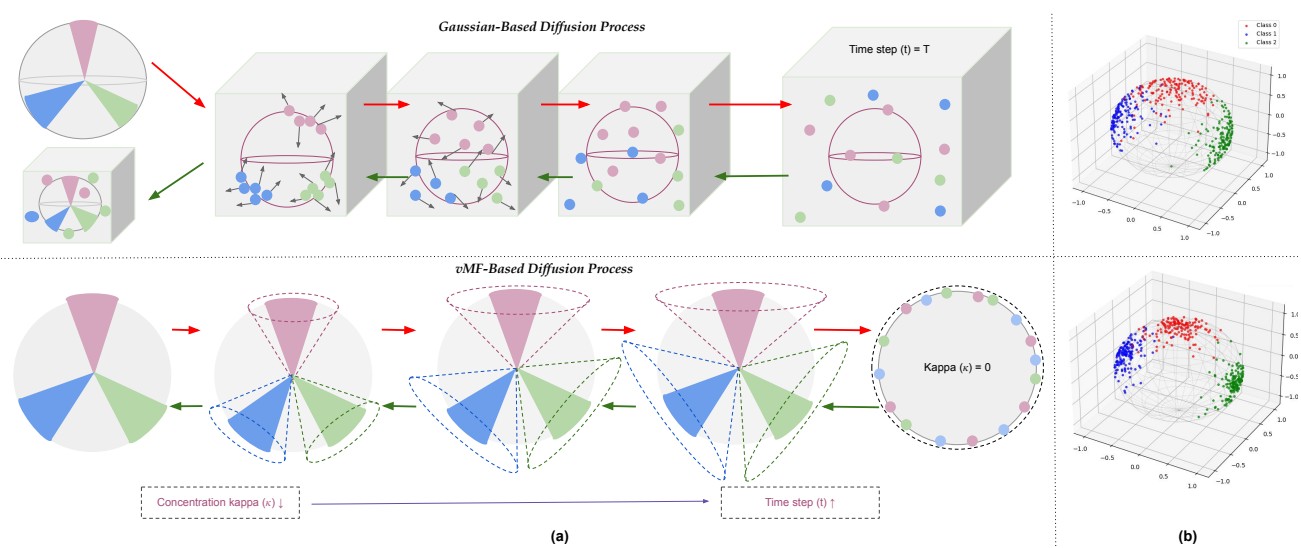

*Figure 1.* (a) Illustrating class geometry preservation in non-Euclidean hyperspherical spaces. The top row shows Gaussian diffusion, which fails to capture structural relations, whereas the bottom row demonstrates von Mises–Fisher (vMF)-based diffusion, effectively modeling directional uncertainty. Red and green arrows indicate forward and backward diffusion process, respectively. (b) Comparison of on the 3D sphere: Gaussian (top) vs. vMF (bottom). Gaussian distorts class boundaries, while vMF maintains the original geometry, preserving angular regions.

ing *directional uncertainty* into the diffusion process. It is visually illustrated through Figure 1(a) (bottom row). Here, **geometry** precisely captures angular relationships defining class boundaries, whereas **uncertainty** reflects stochastic variation governed by the vMF concentration parameter. Figure 1(b) provides empirical visualization demonstrating that vMF-driven diffusion preserves hyper-conical class geometry, while Gaussian-based diffusion distorts it. By embedding vMF-based noise into both forward and reverse processes, the manifold-aware framework maintains global structure and local directional fidelity. It ensures directional uncertainty respects hyperspherical structures throughout the diffusion process. The key contributions of this research are:

1. **Class-Specific *Hypercone* Formalism.** We introduce a hypercone representation that preserves intra-class angular concentration on the hypersphere while enforcing inter-class separation. This formalism captures class structure by focusing on directional relationships rather than purely Euclidean distances.

2. **vMF-Based Forward and Reverse Processes.** Gaussian noise is replaced with vMF noise to maintain angular consistency. This ensures generated samples remain on the hypersphere, effectively guiding them toward class-specific hypercones during the reverse process.

3. **Diverse and Hard Sample Generation.** In contrast to Gaussian-based methods, the vMF-driven approach mitigates simplicity bias, producing a broader sample

distribution. We introduce two metrics, *Hypercone Coverage Ratio* (HCR) and *Hypercone Difficulty Skew* (HDS), for assessing geometry preservation.

4. **Theoretical Foundations and Empirical Validation.** Along with detailed theoretical foundations, through extensive experiments on four object datasets and two face datasets, we demonstrate improved alignment with hyperspherical geometry. We observed enhanced robustness to varying uncertainty levels and superior performance in class-conditional generation tasks.

## 1.2. Related Work

Denoising diffusion models can generate diverse unseen image samples by training on existing datasets (Ho et al., 2020; Song et al., 2021; Dhariwal & Nichol, 2021). First introduced by (Sohl-Dickstein et al., 2015), they added Gaussian noise in a forward process and learned to reverse it for sample generation. Advances in score-based modeling (Song & Ermon, 2019; Pidstrigach, 2022), efficient sampling (Watson et al., 2022), and discrete diffusion (Austin et al., 2021) have further improved their capabilities. However, their reliance on Gaussian noise limits them to Euclidean spaces, restricting their ability to model directional relationships in non-Euclidean domains (Bronstein et al., 2017). While manifold-aware and sphere-based extensions (Rezende et al., 2020) attempt to address this, they remain limited in handling anisotropic and angular uncertainties.

The von Mises-Fisher (vMF) distribution, commonly used for hyperspherical data, has demonstrated effectiveness in

face recognition (Hasnat et al., 2017; Deng et al., 2019), outlier detection and generation (Du et al., 2022; Ming et al., 2023; Du et al., 2023) and representation learning tasks (Davidson et al., 2018). It has also been employed in generative modeling through hyperspherical GANs (Davidson et al., 2018) and spherical VAEs (Falorsi et al., 2018). However, the integration of vMF noise into diffusion models remains relatively unexplored. Previous works exploring non-Gaussian noise in diffusion processes (Wang et al., 2022; Xu et al., 2023) lack a systematic approach to leverage angular distributions like vMF.

In addition to Gaussian and vMF noise, alternative noise types such as Laplacian noise (Dwork & et al., 2006), Student's t-noise (Matsubara & Imai, 2021), and blue noise (Huang et al., 2024) have been explored for robustness and outlier handling. Flow-based transformations have also been used in generative tasks (Kingma & Dhariwal, 2018), while domain-specific noises in molecular generation (Luo et al., 2021) focus on aligning noise with data properties. Our work adds to this growing body of literature by introducing a novel angular noise mechanism based on vMF distributions, enabling the diffusion process to model class-specific and directional uncertainty in hyperspherical domains. Progressive Distillation (Salimans & Ho, 2022) uses an angular DDIM update in Euclidean space but ignores hyperspherical constraints. EDM (Karras et al., 2022) ensures variance preservation via dataset-dependent scaling but lacks directional or geometric alignment. In contrast, *HyperSphereDiff* operates directly on the hypersphere, preserving both variance and manifold consistency without extrinsic normalization.

## 2. Preliminary

In Gaussian-based diffusion models, the forward process is defined as a Markov chain that incrementally corrupts the data by introducing isotropic Gaussian noise. Let $\mathbf{x}_0 \sim p_{\text{data}}(\mathbf{x}_0)$ represent the original data. The sequence of noisy samples $\mathbf{x}_t$ is generated according to:

$$\mathbf{x}_t = \sqrt{\bar{\alpha}_t}\mathbf{x}_0 + \sqrt{1 - \bar{\alpha}_t}\boldsymbol{\epsilon},$$

where, $\alpha_t \in (0, 1]$ is a variance schedule controlling the relative weight of the signal and noise at time $t$ and $\boldsymbol{\epsilon} \sim \mathcal{N}(\mathbf{0}, \mathbf{I})$ is isotropic Gaussian.

The geometry of the corrupted data distribution in a time step $t$ is defined by the weighted combination of the original data $\mathbf{x}_0$ and the noise component $\boldsymbol{\epsilon}$. The resulting distribution can be expressed as:

$$q(\mathbf{x}_t|\mathbf{x}_0) = \mathcal{N}(\mathbf{x}_t; \sqrt{\bar{\alpha}_t}\mathbf{x}_0, (1 - \bar{\alpha}_t)\mathbf{I}),$$

which defines a trajectory in the Euclidean space where the corrupted data progressively transitions from the original

distribution $p_{\text{data}}(\mathbf{x}_0)$ to a standard Gaussian distribution $\mathcal{N}(\mathbf{0}, \mathbf{I})$ as $t \to T$.

The uncertainty in the forward process is introduced through the isotropic Gaussian noise $\boldsymbol{\epsilon} \sim \mathcal{N}(\mathbf{0}, \mathbf{I})$. At each time step, the variance of the noise component $1 - \alpha_t$ increases as $\alpha_t$ decreases, representing a controlled diffusion of information. This uncertainty is modeled symmetrically across all dimensions of the data, resulting in a spherically symmetric probability density function. Formally, the marginal distribution at time $t$ can be expressed as:

$$q(\mathbf{x}_t) = \int q(\mathbf{x}_t|\mathbf{x}_0)p_{\text{data}}(\mathbf{x}_0)\,d\mathbf{x}_0,$$

which represents the corrupted data distribution as a Gaussian mixture where each component is centered at $\sqrt{\bar{\alpha}_t}\mathbf{x}_0$ and has variance $1 - \bar{\alpha}_t$. The reverse process is designed to approximate $p(\mathbf{x}_0|\mathbf{x}_t)$ and recover the original data by progressively denoising the sample $\mathbf{x}_t$.

## 3. Revisiting Hyperspherical Data Geometry

The use of the Gaussian noise simplifies theoretical analysis and computational implementation by assuming data resides in the flat, unbounded Euclidean space $\mathbb{R}^d$. However, this approach is fundamentally sub-optimal when dealing with non-Euclidean data geometries, such as hyperspheres $\mathbb{S}^{d-1}$, or other manifolds.

---

**Theorem 3.1** (Gaussian Noise and Spherical Structure). *Let* $\mathbf{z} \sim \mathcal{N}(\mathbf{0}, \mathbf{I})$ *be an isotropic Gaussian in* $\mathbb{R}^d$. *Then:*

*(a) The radial density follows:*

$$P(\|\mathbf{z}\| = r) \propto r^{d-1}e^{-r^2/2}$$

*(b) As* $d \to \infty$:

$$\frac{\|\mathbf{z}\|}{\sqrt{d}} \xrightarrow{P} 1$$

*(c) For data* $\mathbf{x} \in \mathbb{S}^{d-1}$, *the noised vector* $\mathbf{x} + \sigma\mathbf{z}$ *does not preserve angular relationships as* $\sigma \to \infty$.

*Proof.* (a) Follows from the Jacobian of spherical coordinates, (b) By the law of large numbers, $\|\mathbf{z}\|^2/d \to 1$ in probability, and (c) As $\sigma \to \infty$, the contribution of $\mathbf{x}$ becomes negligible. $\square$

---

Gaussian noise works well in flat, unbounded spaces due to its isotropic nature. However, many real-world datasets reside on curved spaces, such as hyperspheres or other Riemannian manifolds. In these cases, Gaussian noise disrupts the intrinsic geometry by introducing perturbations that extend beyond the manifold (Bronstein et al., 2017; Huang

et al., 2022) (detailed proof in Appendix A). This misalignment motivates the exploration of alternative noise processes designed for data with non-Euclidean geometries.

Facial data embeddings, often derived through deep neural networks, are typically normalized to lie on a hypersphere $\mathbb{S}^{d-1} \subset \mathbb{R}^d$, reflecting their natural angular variability due to factors like pose, expression, and illumination changes (Majumdar et al., 2017; Wang et al., 2018; Deng et al., 2019). This normalization ensures that the similarity between two embeddings is determined by their angular relationship rather than their magnitude, aligning with the cosine similarity metric frequently used in recognition tasks. For two embeddings $\mathbf{e}_1, \mathbf{e}_2 \in \mathbb{S}^{d-1}$, their similarity can be expressed as $\cos(\theta) = \mathbf{e}_1^\top \mathbf{e}_2$, where $\theta = \arccos(\mathbf{e}_1^\top \mathbf{e}_2)$ is the geodesic distance on the hypersphere. This structure inherently aligns facial embeddings with hyperspherical geometry, where angular deviations are the primary measure of variability. In high-dimensional hyperspherical spaces, facial classes can be modeled as distinct regions, often visualized as hypercones emanating from the origin. Each class $C_k \subset \mathbb{S}^{d-1}$ is centered around a mean direction vector $\boldsymbol{\mu}_k \in \mathbb{S}^{d-1}$, with intra-class variability characterized by angular deviations. The vMF distribution provides a natural framework for modeling such hypercones due to its concentration parameter $\kappa_k$, which controls the spread of the distribution around $\boldsymbol{\mu}_k$. The probability density function is given as:

$$f(\mathbf{x}; \boldsymbol{\mu}, \kappa) = \frac{\kappa^{(d/2)-1}}{(2\pi)^{d/2} I_{(d/2)-1}(\kappa)} \exp(\kappa \boldsymbol{\mu}^\top \mathbf{x}),$$

where, $I_{(d/2)-1}(\kappa)$ is the modified Bessel function of the first kind. To formalize this relationship, we introduce the following lemma:

---

**Lemma 3.2** (vMF Hypercone Representation). *Let $\mathcal{C}_k \subset \mathbb{S}^{d-1}$ be a class hypercone centered at $\boldsymbol{\mu}_k$ with angular radius $\theta_k$. The data follows a vMF distribution with concentration $\kappa_k$. Then:*

*(a) The probability mass within the class hypercone is:*

$$P(\mathbf{x} \in \mathcal{C}_k) = P(\angle(\mathbf{x}, \boldsymbol{\mu}_k) \leq \theta_k) \geq 1 - \exp(-\kappa_k(1 - \cos\theta_k))$$

*(b) For any desired coverage probability $1 - \epsilon$ with $\epsilon \in (0, 1)$, choosing:*

$$\kappa_k \geq \frac{1}{1 - \cos\theta_k} \log\left(\frac{1}{\epsilon}\right)$$

*ensures that $P(\mathbf{x} \in \mathcal{C}_k) \geq 1 - \epsilon$.*

***Note:*** *As $\theta_k \to 0$, the required concentration $\kappa_k \to \infty$, reflecting the scenario of a single-direction hypercone.*

---

On the hypersphere $\mathbb{S}^{d-1}$, classes naturally align within hypercones defined by angular constraints, and the vMF distribution inherently models these structures. Such distributions also relate to class separation on hyperspheres (refer Appendix B). Unlike Gaussian noise, vMF noise respects the manifold's geometry by modulating its focus through the concentration parameter $\kappa$. When $\kappa = 0$, vMF is isotropic, uniformly spanning the hypersphere, while larger $\kappa$ values focus the distribution within a hypercone around the mean direction $\boldsymbol{\mu}$. This adaptability enables vMF-based diffusion to effectively learn and represent class-wise structures, ensuring a geometrically consistent generative modeling framework.

## 4. Modeling Angular Uncertainty in Diffusion

To effectively model the uncertainty inherent in hyperspherical data, we employ a forward diffusion process driven by noise sampled from the vMF distribution. The forward process introduces angular-based uncertainty such that the data progressively transitions from structured, class-specific noise to a uniform distribution over the hypersphere $\mathbb{S}^{d-1}$. Initially, the noise is highly concentrated ($\kappa$ is large), preserving the class structure within narrow hypercones. As the process advances, $\kappa$ decreases to zero, injecting isotropic noise, and the data diffuses uniformly over the hypersphere.

### 4.1. Angular Noise Injection

In the proposed *HyperSphereDiff*, the forward process adds noise to the data representation using angular interpolation. At each time step $t$, the data representation obtained in a latent space through an encoder is denoted as $\mathbf{z}_t$, which is updated as $\mathbf{z}_t = \cos(\theta_t)\mathbf{z}_{t-1} + \sin(\theta_t)\mathbf{v}$, where $\mathbf{z}_{t-1} \in \mathbb{S}^{d-1}$ is the data representation at the previous time step, $\mathbf{v}$ is a unit vector sampled uniformly from $\mathbb{S}^{d-1}$, and $\theta_t$ is a time-dependent angle that increases monotonically from $0$ to $\pi/2$. The angular interpolation ensures that the injected noise aligns with the hyperspherical geometry. The parameter $\theta_t$ acts as a scheduler, gradually increasing to control the extent of deviation from the previous representation. Correspondingly, this forward step mimics the vMF distribution using $\kappa_t$ as a scheduler. The decrease in $\kappa_t$ defines the level of angular uncertainty introduced at each step.

By employing $\kappa$ as a scheduler, our forward diffusion process progressively incorporates angular uncertainty, transitioning from class-structured perturbations to isotropic noise (Figure 1). This ensures that the initial steps of the corruption process retain class information, as larger $\kappa$ values concentrate the data around class-specific hypercones. Over time, as $\kappa \to 0$, the noise diffuses the data to a uniform hypersphere, reflecting maximum uncertainty. This process inherently respects the geometric properties of hyperspherical data, with angular deviations tied to the data

distribution rather than being purely random.

## 4.2. Backward Step: Hypersphere to Hypercone

In generative modeling with hyperspherical data, the reverse process of *HyperSphereDiff* transforms noisy samples into structured data aligned with class distributions. This process progressively refines noisy points toward class-specific hypercones using score-based methods. The reverse formulation leverages vMF sampling, ensuring angular relationships between classes are retained. The reverse process is defined as:

$$\mathbf{z}_{t-1} \sim \text{vMF}\left(\Pi(\mathbf{z}_t + \eta_t \nabla_{\mathbf{z}_t} \log f(\mathbf{z}_t; \boldsymbol{\mu}_t, \kappa_t)), \kappa_t\right),$$

The complete algorithm A and experiments are based on this reverse step and MSE loss as defined in standard Gaussian diffusion. Alternatively, the reverse step with angular updates using vMF-based stochastic denoising is defined as:

$$z_{t-1} \sim \text{vMF}\Bigg(\Pi\Big(\cos(\theta_t)z_t + \\ \sin(\theta_t)\frac{\nabla_{z_t} \log f(z_t; \mu_c)}{\|\nabla_{z_t} \log f(z_t; \mu_c)\|}\Big), \kappa_t\Bigg)$$

To further align optimization with hyperspherical geometry, the reverse denoising step is refined using an angular-based loss function alternative to the MSE loss. Cosine Loss: Encourages angular alignment between the score function and noise direction.

$$\mathcal{L}c = 1 - \mathbb{E}\left[\nabla z_t \log f(z_t; \mu_c)^\top \epsilon_t\right]$$

Geodesic Loss: Penalizes angular deviations.

$$\mathcal{L}g = \mathbb{E}\left[\arccos^2\left(\frac{\nabla z_t \log f(z_t; \mu_c)^\top \epsilon_t}{\|\nabla_{z_t} \log f(z_t; \mu_c)\|\|\epsilon_t\|}\right)\right]$$

where, $\mathbf{z}_t$ represents the current noisy sample, $f(\cdot)$ is the vMF distribution modeling class hypercones, and $\nabla_{\mathbf{z}_t} \log f(\cdot)$ is the score function providing gradient information. Here, interpolation using $\cos(\theta_t)$ and $\sin(\theta_t)$ maintains angular relationships, while the normalized score function preserves directional consistency. (Refer Appendix C) The comparative analysis of various loss functions and angular based denoising is provided in Appendix (Refer Appendix H.2).

The process evolves from isotropic noise to class-specific structures through the interplay of two key components. The score function $\nabla_{\mathbf{z}_t} \log f(\mathbf{z}_t; \boldsymbol{\mu}_t, \kappa_t)$ points towards the class means $\boldsymbol{\mu}_c$, with the step size $\eta_t$ controlling the update strength. Simultaneously, vMF sampling adds controlled stochasticity, ensuring updates remain consistent with hyperspherical geometry while progressively removing noise.

The concentration parameter $\kappa_t$ plays a crucial role in shaping this evolution. It starts small for isotropic diffusion and then grows during the reverse process, increasingly favoring class-specific directions. As $\kappa_t$ increases, the vMF sampling becomes more concentrated around the class means, reflecting the growing certainty in class membership while maintaining geometric consistency.

The combination of score-based updates and vMF sampling ensures a smooth transition from noise to structured data. The process gradually refines points until they converge near their respective class means $\boldsymbol{\mu}_c$ for a class $c$, effectively recovering true class representations while preserving the underlying hyperspherical geometry. This mechanism provides a mathematically principled approach to generating class-consistent samples on the hypersphere, maintaining both local structure and global class relationships throughout the diffusion process.

---

**Algorithm 1** *HyperSphereDiff Training:* vMF Diffusion with Hypercone Preservation

---

**Require:** Data samples $\{\mathbf{x}_i\}$, class labels $\{y_i\}$, diffusion steps T
**Require:** Angular schedule $\{\theta_t\}_{t=1}^T$, Learning rate $\eta$
 1: Initialize score network parameters $\theta$
 2: **while** not converged **do**
 3:  Sample batch $(\mathbf{x}, y)$ from dataset
 4:  Sample time step $t \sim \text{Uniform}(1, T)$
 5:  $\kappa_t \leftarrow \cot(\theta_t)$ ▷ Set concentration
 6:  Sample $\mathbf{v} \sim \text{Uniform}(\mathbb{S}^{d-1})$
 7:  $\mathbf{z}_t \leftarrow \Pi(\cos(\theta_t)\mathbf{x} + \sin(\theta_t)\mathbf{v})$ ▷ Forward process
 8:  similar to $\mathbf{z}_t \sim \text{vMF}(\mathbf{x}, \kappa_t)$
 9:  $\nabla_{\mathbf{z}_t} \log f \leftarrow \text{ScoreNet}_\theta(\mathbf{z}_t, t, y)$
10:  $\mathcal{L} \leftarrow \|\nabla_{\mathbf{z}_t} \log f - \nabla_{\mathbf{z}_t} \log p(\mathbf{z}_t|\mathbf{x})\|^2$
11:  Update $\theta$ using gradient of $\mathcal{L}$
12: **end while**
13: **return** Trained score network parameters $\theta$

---

## 4.3. Adaptive Class-Dependent Concentration

We consider data $\mathbf{z}_0$ on the unit hypersphere and a forward diffusion process as before with vMF transitions. In the reverse process $p_\theta(\mathbf{z}_{0:T}) = p_\theta(\mathbf{z}_0 \mid \mathbf{z}_1, y) \prod_{t=1}^T p_\theta(\mathbf{z}_{t-1} \mid \mathbf{z}_t, y)$, we guide samples into learned truncated regions on the hypersphere. We employ two networks: a direction predictor $D_\phi$ that estimates class-specific directions $\mathbf{m}_t = D_\phi(\mathbf{z}_t, t, y)$, and an angle predictor $C_\psi$ that estimates class angular radii $\theta_y = C_\psi(\mathbf{z}_t, t, y)$. These networks learn to form dynamic hypercones $\mathcal{C}_{t,y} = \{\mathbf{z} : \angle(\mathbf{z}, \mathbf{m}_t) \leq \theta_y\}$ that capture class-specific regions at each step. The reverse process uses truncated vMF:

$$p_\theta(\mathbf{z}_0 \mid \mathbf{z}_1, y) = \text{TvMF}\big(\mathbf{z}_0; \mathbf{m}_\theta(\mathbf{z}_1, y), \kappa_\theta(\mathbf{z}_1, y), \mathcal{C}_{t,y}\big),$$

**Algorithm 2** *HyperSphereDiff Testing:* Sampling from vMF Diffusion with Class Guidance

**Require:** Class label $y$, diffusion steps $T$, trained score network $\theta$
**Require:** Angular schedule $\{\theta_t\}_{t=1}^T$, step sizes $\{\eta_t\}_{t=1}^T$
1: Sample $\mathbf{z}_T \sim \text{Uniform}(\mathbb{S}^{d-1})$
2: **for** $t = T$ to 1 **do**
3:     $\kappa_t \leftarrow \cot(\theta_t)$           $\triangleright$ Get concentration
4:     **if** $t > 1$ **then**
5:         $\nabla_{\mathbf{z}_t} \log f \leftarrow \text{ScoreNet}_\theta(\mathbf{z}_t, t, y)$
6:         $\mathbf{m}_t \leftarrow \mathbf{z}_t + \eta_t \nabla_{\mathbf{z}_t} \log f$
7:         $\mathbf{m}_t \leftarrow \mathbf{m}_t / \|\mathbf{m}_t\|$     $\triangleright$ Project to hypersphere
8:         $\mathbf{z}_{t-1} \sim \text{vMF}(\mathbf{m}_t, \kappa_t)$
9:     **end if**
10: **end for**
11: **return** Final sample $\mathbf{z}_1$

$$\text{TvMF}(\mathbf{z}; \mu, \kappa, \mathcal{C}) = \frac{1}{Z(\kappa, \mathcal{C})} \exp(\kappa \, \mu^\top \mathbf{z}) \, \mathbf{1}\{\mathbf{z} \in \mathcal{C}\},$$

We maintain adaptive concentration through $\kappa_\theta(\mathbf{z}_t, y) = \kappa_{\max} \sigma\big(\beta \left[\theta_y - \angle(\mathbf{z}_t, \mathbf{m}_t)\right]\big)$, where $\beta$ is a scaling factor, $\kappa_{max}$ is maximum concentration allowed and $\sigma(.)$ is sigmoid function, ensuring stronger concentration within predicted regions. The normalization constant $Z(\kappa, \mathcal{C})$ for the truncated vMF can be computed as:

$$Z(\kappa, \mathcal{C}) = \int_{\mathcal{C}} \exp(\kappa \, \mu^\top \mathbf{z}) \, d\mathbb{S}^{d-1}(\mathbf{z})$$

where, $d\mathbb{S}^{d-1}$ is the hyperspherical measure. This integral has a closed form in terms of the incomplete gamma function when $\mathcal{C}$ is a hypercone. The networks $D_\phi$ and $C_\psi$ are trained jointly with the score network to minimize the objective:

$$\mathcal{L}(\phi, \psi) = \mathbb{E}_{t,y} \left[ \|\mathbf{m}_t - \hat{\mathbf{m}}_y\|^2 + \lambda(\theta_y - \hat{\theta}_y)^2 \right]$$

where, $\hat{\mathbf{m}}_y$ and $\hat{\theta}_y$ are empirical class statistics computed from training data, and $\lambda$ balances the direction and angle losses. This ensures that the predicted geometry aligns with the true class structure while maintaining the flexibility of learned truncation regions.

The combination of learned directional prediction, adaptive concentration, and explicit truncation provides a powerful mechanism for class-conditional generation. At each step $t$, the process maintains a dynamic balance between score-based updates and geometric constraints:

$$\mathbb{E}[\mathbf{z}_{t-1}|\mathbf{z}_t, y] = \mathbf{m}_t + \eta_t \nabla_{\mathbf{z}_t} \log f(\mathbf{z}_t; \mu_t, \kappa_t) \cdot \mathbf{1}\{\mathbf{z}_t \in \mathcal{C}_{t,y}\}$$

This ensures samples remain within class-appropriate regions while benefiting from the score function's gradient information. Detailed proofs of forward and reverse modeling are shown in Appendix D.

*Table 1.* Performance comparison of Gaussian and *HyperSphereDiff* across six datasets using FID (lower is better), HCR (lower is better), and HDS (lower values indicates harder samples). The vMF model demonstrates superior capability in generating challenging samples with better FID and HDS scores.

| Dataset | FID | | HCR | | HDS | |
|---|---|---|---|---|---|---|
| | Gaussian | vMF | Gaussian | vMF | Gaussian | vMF |
| MNIST | 1.95 | 1.86 | 0.17 | 0.14 | 0.76 | 0.52 |
| CIFAR-10 | 3.45 | 3.52 | 0.23 | 0.2 | 0.72 | 0.48 |
| CUB-200 | 8.47 | 9.11 | 0.19 | 0.13 | 0.81 | 0.41 |
| Cars-196 | 9.09 | 7.87 | 0.19 | 0.17 | 0.85 | 0.6 |
| CelebA | 9.31 | 9.29 | 0.42 | 0.22 | 0.77 | 0.59 |
| D-LORD | 11.38 | 9.27 | 0.46 | 0.21 | 0.91 | 0.62 |

### 4.4. Beyond Geometry and Uncertainty: Retaining Image Information

In our face generation process, we introduce two forms of uncertainty to capture both *magnitude* and *directional* components in the data representation motivated by (Chiranjeev et al., 2024). Specifically, we interpret an image embedding $\mathbf{x}$ as having a *magnitude* $\|\mathbf{x}\|$ (which relates to overall intensity or scale) and a *direction* $\mathbf{x}/\|\mathbf{x}\| \in \mathbb{S}^{d-1}$ (which captures structural and class-related geometry). We employ a Gaussian distribution to model the magnitude uncertainty and a vMF distribution to model the directional uncertainty. This hybrid approach preserves crucial *class geometry* on the hypersphere, while still accommodating image-specific variability (e.g. changes in lighting or intensity) through Gaussian noise. The detailed forward and reverse process is provided in Appendix F.

## 5. Experiments

We evaluate *HyperSphereDiff* on four object datasets (CIFAR-10 (Krizhevsky et al., 2009), MNIST (Deng, 2012), CUB-200 (Wah et al., 2011) and Cars-196 (Krause et al., 2013)), and two face datasets (CelebA (Liu et al., 2015) and D-LORD (Manchanda et al., 2023)). The architecture and training details are provided in Appendix H.

**Retaining Geometry:** We define two metrics to evaluate whether the generated samples preserve the class structure and maintain distributional properties of the original data.
**Hypercone Coverage Ratio (HCR):** The HCR quantifies the percentage of generated samples outside the class distribution's hypercone. For each class k, we compute:

$$\text{HCR} = \frac{1}{K} \sum_{k=1}^K \frac{1}{N_k} \sum_{i=1}^{N_k} \mathbb{1}\left[ \cos^{-1}(\mathbf{z}_i \cdot \mu_k) > \theta_k^{\max} \right]$$

A lower HCR indicates better preservation of the class structure, while a higher HCR suggests out-of-class or unrealistic samples.
**Hypercone Difficulty Skew (HDS):** The HDS measures whether the model generates easy samples by analyzing how

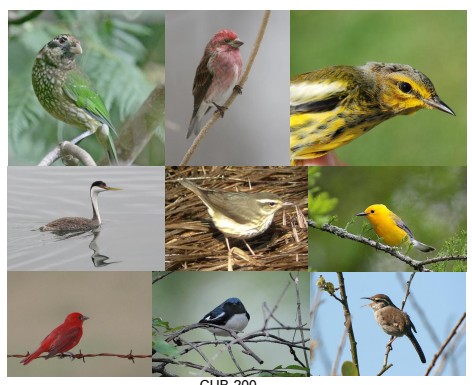 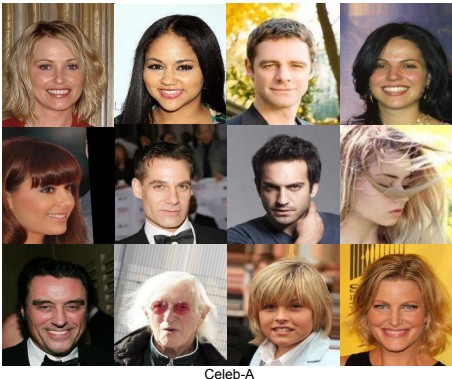 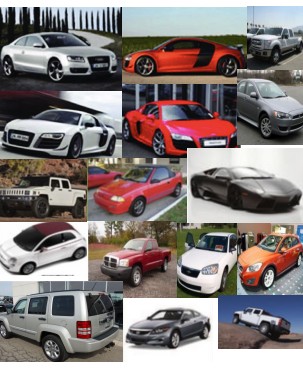

CUB-200             Celeb-A             Cars196

*Figure 2.* Generated samples of birds (left), faces (middle) and cars (right) using the proposed vMF based diffusion model trained on CUB-200, CelebA and Cars-196 dataset, showcasing the preservation of key attributes and diversity.

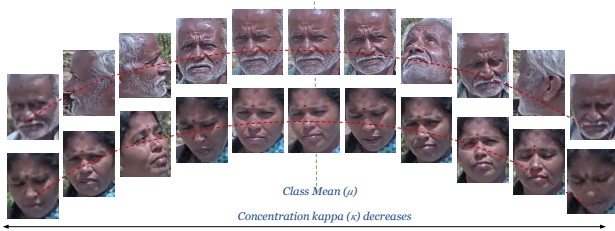

*Class Mean (μ)*

*Concentration kappa (κ) decreases*

*Figure 3.* Real-world surveillance samples generated using class-dependent adaptive concentration $\kappa$, through proposed diffusion model after training on D-LORD dataset.

they are distributed across sub-cones of increasing angular deviation. A high HDS indicates the model generates easy samples, while a low HDS suggests a balanced distribution across difficulty levels (refer Appendix H.1). For each class $k$, we compute the fraction of samples in each sub-cone and compute HDS:

$$p_k^m = \frac{1}{N_k} \sum_{i=1}^{N_k} \mathbb{1}\left[\theta_{m-1} < \cos^{-1}(\mathbf{z}_i \cdot \boldsymbol{\mu}_k) \leq \theta_m\right]$$

$$\text{HDS} = \frac{1}{K} \sum_{k=1}^{K} \sum_{m=1}^{M} w_m p_k^m$$

The HCR and HDS metrics offer a thorough assessment of generative models in angular space. HCR measures class consistency and assesses how well intra-class variations are preserved, while HDS identifies whether the model favors generating easier samples. An ideal generative model should maintain a low HCR while achieving an HDS that aligns with the natural difficulty distribution of the original data.

Table 1 highlights the comparison between the Gaussian and vMF models in terms of HCR and HDS in six datasets. The Hypercone Coverage Ratio (HCR) consistently decreases for the vMF model compared to the Gaussian model, indicat-

ing that the vMF generates samples that follow the original class structure, thus preserving angular relations. Similarly, the Hypercone Difficulty Skew (HDS) value with the vMF model in all six datasets is around $0.5$, reflecting its ability to generate generalized samples across the hypercone rather than generating simple samples. Gaussian-based generated samples have HDS values around $0.6$, showing that a higher proportion of samples are generated from the innermost hypercone demonstrating simplicity bias. For datasets like D-LORD and CUB-200, the reduction in HCR and HDS is particularly pronounced, showcasing the efficiency of the vMF model in modeling angular variation and producing diverse samples.

**Generation Quality:** We evaluate the quality of generated samples using our method across six diverse datasets. These include natural images (CelebA, CIFAR-10), fine-grained object categories (CUB-200, Cars-196), and structured digits (MNIST). Figure 2 presents examples of generated faces and birds from models trained on CelebA, Cars-196, and CUB-200. The images exhibit realistic structural coherence and diversity, resembling their respective distributions. We also compute the Fréchet Inception Distance (FID) (Bynagari, 2019), achieving competitive scores across all datasets (Table 1). In particular, CelebA, D-LORD, and Cars-196 show high performance, indicating that the model captures both global structure and fine-grained details. This demonstrates that *HyperSphereDiff* generates high-quality samples with significant variation while preserving class structure, aligning with our theoretical insights (See Appendix H).

**Hypercone Generation:** We evaluate the hypercone constraint by analyzing generated samples from both inner and outer hypercones, shown in Appendix H. Samples in the inner hypercone retain class-consistent features, while those in the outer hypercone exhibit noisy generations, validating the effectiveness of our metric in capturing generation quality. Furthermore, for the real-world D-LORD (Manchanda

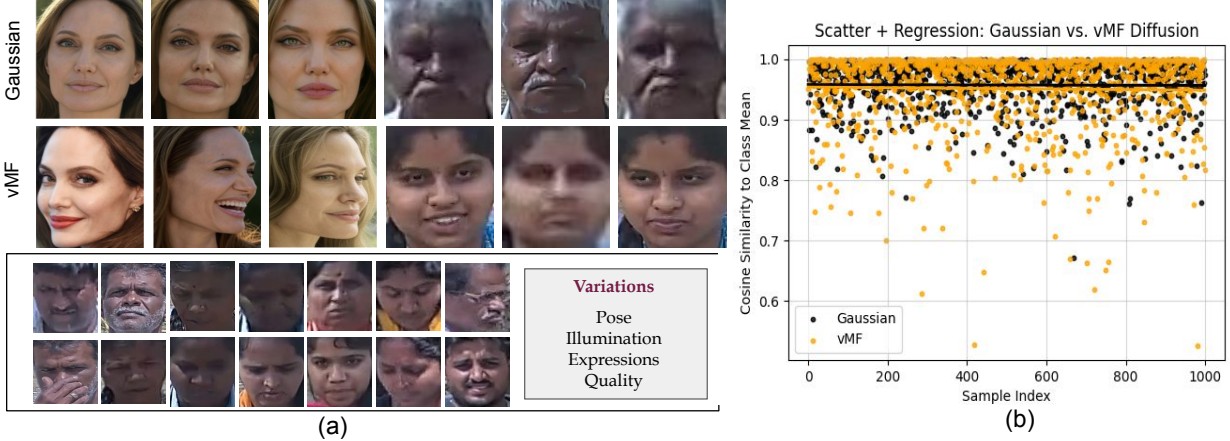

(a)                                          (b)

*Figure 4.* (a) Comparison of samples generated using Gaussian and vMF diffusion models, highlighting improved variation in pose, illumination, expression, and quality with vMF. (b) Scatter plot with fitted regression shows cosine similarity to the class mean, where *HyperSphereDiff* generates more challenging samples with broader similarity distribution compared to Gaussian.

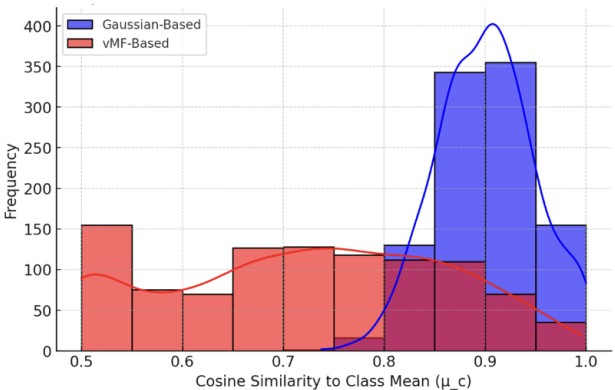

*Figure 5.* Gaussian-based diffusion generates samples tightly clustered near the class mean (high similarity, low variance), favoring easier cases. In contrast, *HyperSphereDiff* produces a more diverse spread (lower similarity, higher variance), ensuring better coverage across difficulty levels.

*Table 2.* FID scores for different noise configurations across Celeb-A and D-LORD datasets. The hybrid Gaussian + Spherical method achieves the best performance.

| Dataset | Gaussian | Gaussian + Spherical | Spherical |
|---------|----------|----------------------|-----------|
| Celeb-A | 9.31 | **9.29** | 9.31 |
| D-LORD | 11.38 | **9.27** | 10.94 |

*iff* compared to the Gaussian model. This result highlights the ability of our approach to generate data that better aligns with the angular structure of real-world surveillance images, effectively capturing the underlying geometry.

**Generating Hard Samples:** We observed that the vMF-based approach consistently produced samples with varied cosine similarity to the class mean, indicating a higher degree of generalized sample generation compared to the Gaussian baseline, as shown in Figure 4. This is also supported by qualitative results, where vMF samples exhibit greater variations in pose, illumination and expression, closely resembling real-world challenges. Quantitatively, this observation is validated using HDS metric (Table 1), which measures the proportion of samples concentrated in the smaller hypercones. The vMF model achieved lower HDS values, confirming its effectiveness in generating harder, more diverse samples that can better represent challenging scenarios in real-world datasets. Further, Figure 5 shows *HyperSphereDiff* yields broader coverage (mean cosine similarity=0.722, std=0.137), reflecting balanced difficulty representation compared to Gaussian diffusion (mean=0.900, std=0.048).

**Ablation Study:** Table 2 presents FID scores comparison on Celeb-A and D-LORD datasets under three noise strategies: Gaussian, Spherical (*HyperSphereDiff*), and a hybrid of

et al., 2023) dataset, we sample images at varying kappa values shown in Figure 3, concentrating the samples around the mean vector to reflect distance-wise variations in the surveillance imagery. These experiments highlight how diffusion with hypercone constraints ensures both intra-class consistency and controlled diversity.

**Face Recognition Results:** We evaluate the performance of our proposed method on the real-world D-LORD (Manchanda et al., 2023) surveillance dataset, which consists of sequential frame images exhibiting angular relationships in the feature space. We generated samples of 1000 synthetic subjects using both Gaussian-based diffusion and our vMF-based hyperspherical diffusion. Training the ArcFace model on the generated data leads to 5.0% improvement on the real test set identification accuracy with the *HyperSphereD-*

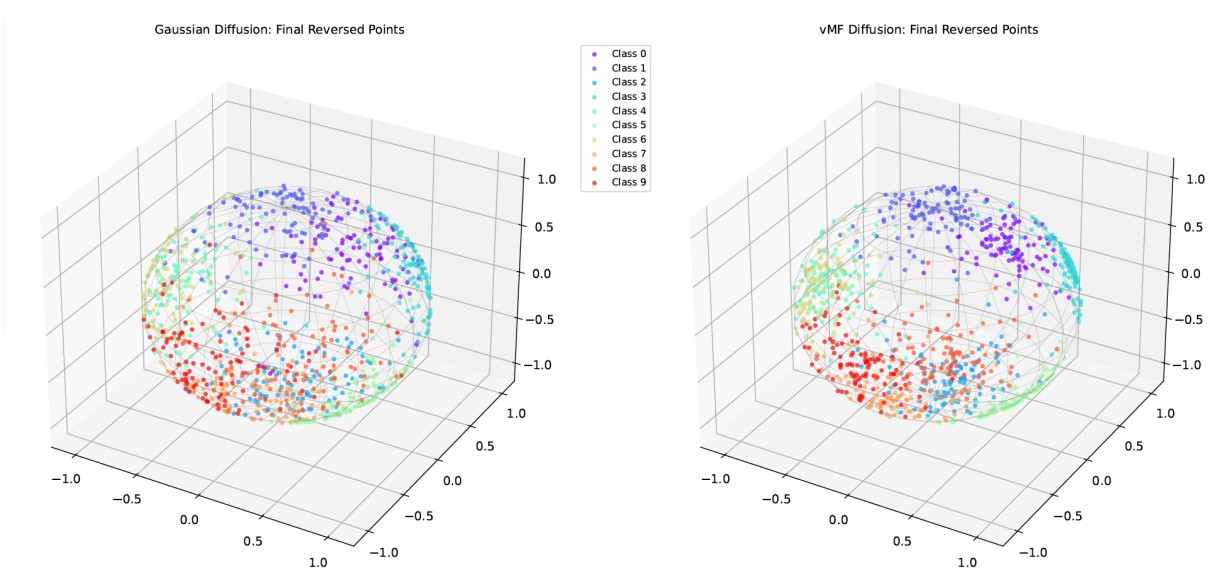

*Figure 6.* Feature representation of the 10-class MNIST dataset generated using Gaussian-based diffusion (left) and vMF-based diffusion (right). The vMF-based sampling aligns generated sample features within class-specific 3D hypercones, while Gaussian-based sampling results in scattered features outside the class-hypercones.

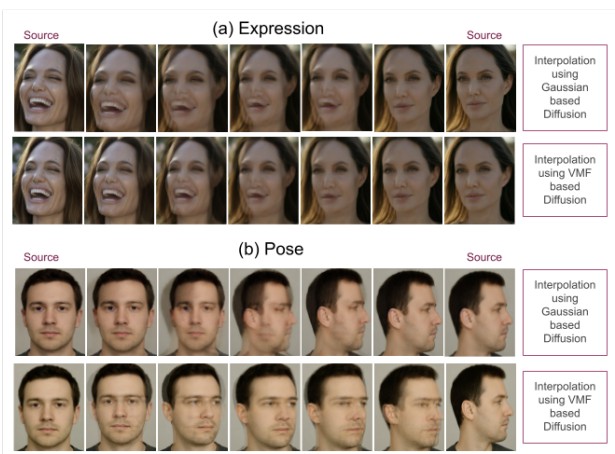

*Figure 7.* Comparison between interpolation using Gaussian-based diffusion and vMF-based diffusion for generating images between two variants of the same subject: (a) Expression, (b) Pose.

both. The hybrid model yields the lowest FID scores across both datasets (9.29 for Celeb-A and 9.27 for D-LORD), suggesting that combining magnitude-based Gaussian noise with direction-aware spherical noise results in better generation quality. Purely Gaussian or purely spherical models alone are less effective, particularly for D-LORD, where Gaussian diffusion performs poorly (FID = 11.38) and the spherical-only model underperforms compared to the hybrid. This demonstrates the advantage of modeling dual uncertainty by capturing both intensity variation and directional structure to enhance the realism and class consistency.

**Feature Representation:** Figure 6 illustrates the feature representations of conditional samples generated from the 10 classes of the MNIST dataset. The figure highlights that the vMF-based reverse sampling effectively converges samples within class-specific hypercones, capturing the angular geometry of the data. In contrast, Gaussian-based reverse sampling produces samples that converge within Euclidean space, failing to adhere to the hyperspherical structure.

**Interpolation Results:** Figure 7 shows interpolations using Gaussian-based diffusion (top row) which often produce unnatural or inconsistent transitions, especially with large attribute shifts. In contrast, our *HyperSphereDiff* (bottom row) angular interpolation on a hypersphere yields smoother, identity-preserving transitions across poses and expressions.

# 6. Conclusion and Future Work

In this work, we have challenged the ubiquitous Gaussian assumption in diffusion models by introducing a novel hyperspherical framework *HyperSphereDiff* that leverages vMF distributions for generative modeling on angular manifolds. By incorporating hyperspherical geometry and class-dependent uncertainty, our approach preserves angular structure while producing diverse, semantically rich samples. Extensive experiments on facial and complex object datasets demonstrate its effectiveness in fine-grained tasks where angular relationships are critical. This work opens new avenues for manifold-constrained generative modeling, advancing geometry-aware diffusion techniques. Future research will focus on developing adaptive $\kappa$-based schedulers, adopting hierarchical hypercone partitioning for finer class variations, and extending the framework to conditional generation tasks, such as pose-invariant face synthesis.

## Impact Statement

This work introduces a novel hyperspherical diffusion framework leveraging von Mises–Fisher (vMF) distributions to enhance the modeling of high-dimensional angular data in Machine Learning. By improving the fidelity and interpretability of generative models, the proposed method has applications in computer vision, fine-grained classification, and surveillance. However, like other generative techniques, it carries potential ethical risks, including misuse in deepfakes, privacy concerns, and bias amplification. To mitigate these risks, we emphasize responsible use and transparency, particularly in sensitive domains (Mittal et al., 2024). Overall, this research advances generative modeling for hyperspherical data while promoting a deeper understanding of geometric structures in Machine Learning.

## Acknowledgements

The authors acknowledge the support of IndiaAI and Meta through the Srijan: Centre of Excellence for Generative AI.

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

## A. Gaussian Noise Distorts Angular Relationship

A key challenge in hyperspherical data modeling is preserving the angular relationships that define class structure, especially when noise is introduced. In many generative and transformation-based approaches, Gaussian noise is commonly used to perturb data points. However, in non-Euclidean spaces like the hypersphere, such noise can significantly distort the underlying geometric structure. Unlike structured perturbations that respect the manifold's constraints, isotropic Gaussian noise introduces deviations that shift points off the sphere and disrupt their relative angles. The following lemma formally establishes how Gaussian noise fails to maintain angular class structure, particularly in high-dimensional spaces where its effects become more pronounced.

---

**Lemma A.1** (Gaussian Noise Distorts Angular Class Structure)**.** *Let $\{\mathbf{x}_i\}_{i=1}^N$ be $N$ points in $\mathbb{S}^{d-1}$ (grouped into classes) such that the* angular distances *(or equivalently, dot products) between points reflect some inter-class and intra-class relationships (e.g. points in the same class are close in angle, points in different classes have larger angles). Define the perturbed points*

$$\widetilde{\mathbf{x}}_i \;=\; \mathbf{x}_i \;+\; \boldsymbol{\epsilon}_i, \quad \boldsymbol{\epsilon}_i \sim \mathcal{N}(\mathbf{0},\,\sigma^2 \mathbf{I}), \quad i = 1,\dots,N,$$

*where, $\boldsymbol{\epsilon}_i$ are i.i.d. isotropic Gaussians in $\mathbb{R}^d$. Then in general, the inner products*

$$\widetilde{\mathbf{x}}_i^\top \widetilde{\mathbf{x}}_j \quad and \quad \mathbf{x}_i^\top \mathbf{x}_j$$

*are* not *preserved; i.e. with high probability, the angles among the perturbed points $(\widetilde{\mathbf{x}}_1,\dots,\widetilde{\mathbf{x}}_N)$ are significantly different from those among the original points $(\mathbf{x}_1,\dots,\mathbf{x}_N)$, especially as $d$ grows and/or $\sigma > 0$ is large.*

***Proof (Sketch):***

i) ***No longer on the hypersphere.*** *Since $\|\boldsymbol{\epsilon}_i\| \neq 0$ almost surely, the perturbed points $\widetilde{\mathbf{x}}_i$ will* not *lie on $\mathbb{S}^{d-1}$, so angles measured in $\mathbb{R}^d$ w.r.t. the origin are already changed. More precisely,*

$$\|\widetilde{\mathbf{x}}_i\|^2 \;=\; \|\mathbf{x}_i + \boldsymbol{\epsilon}_i\|^2 \;\approx\; 1 \;+\; \|\boldsymbol{\epsilon}_i\|^2 \;+\; 2\,(\mathbf{x}_i^\top \boldsymbol{\epsilon}_i),$$

*which (with probability 1) is not equal to 1. Thus, any "spherical" relationships are broken immediately.*

ii) ***Inner products become random.*** *Consider the inner product*

$$\widetilde{\mathbf{x}}_i^\top \widetilde{\mathbf{x}}_j \;=\; (\mathbf{x}_i + \boldsymbol{\epsilon}_i)^\top (\mathbf{x}_j + \boldsymbol{\epsilon}_j) \;=\; \mathbf{x}_i^\top \mathbf{x}_j \;+\; \mathbf{x}_i^\top \boldsymbol{\epsilon}_j \;+\; \mathbf{x}_j^\top \boldsymbol{\epsilon}_i \;+\; \boldsymbol{\epsilon}_i^\top \boldsymbol{\epsilon}_j.$$

*Since $\boldsymbol{\epsilon}_i, \boldsymbol{\epsilon}_j$ are Gaussians with mean 0, each cross-term is a random variable whose distribution depends on $\sigma^2$ and $d$.*

iii) ***High dimension amplifies distortion.*** *In high $d$ with $\sigma^2 > 0$, we typically have $\|\boldsymbol{\epsilon}_i\| \approx \sqrt{d}\,\sigma$, so the energy in the noise vectors can overshadow the original norm ($\|\mathbf{x}_i\| = 1$). Hence $\|\widetilde{\mathbf{x}}_i\| \approx \sqrt{d}\,\sigma$, dominating any small angular differences that originally existed among the $\{\mathbf{x}_i\}$. Even for moderate $d$, if $\sigma$ is large enough, $\widetilde{\mathbf{x}}_i$ and $\widetilde{\mathbf{x}}_j$ become nearly orthogonal or randomly oriented (depending on the sign and correlation among the noise). Thus, the* relative angles *among the perturbed points often bear little resemblance to the original class structure.*

iv) ***Conclusion.*** *Because isotropic Gaussian noise in $\mathbb{R}^d$ shifts points off the hypersphere and injects random directions at scale $\sigma$, it fails to* preserve *the original spherical relationships (both inter-class angles and intra-class distributions). In fact, for large $d$ or sufficiently large $\sigma$, the new angles are effectively random, destroying the class separation that was originally encoded in angles on $\mathbb{S}^{d-1}$.*

---

## B. Class Separation using vMF

A fundamental challenge in generative modeling on hyperspheres is ensuring class separability while preserving the underlying geometric structure. The von Mises-Fisher (vMF) distribution provides a natural way to model directional data while maintaining angular coherence. Unlike Gaussian noise, which distorts class boundaries by introducing random

perturbations in Euclidean space, vMF-based modeling enforces directional consistency by concentrating samples around a mean direction with a tunable spread controlled by the concentration parameter $\kappa$.

The following lemma establishes a probabilistic bound on class separability when data points from different classes are modeled using vMF distributions. It quantifies how the probability of misclassification depends on $\kappa$ and the angular separation $\theta$ between class centers. This result provides a theoretical foundation for setting $\kappa$ to achieve a desired classification accuracy and demonstrates that stronger concentration (higher $\kappa$) exponentially improves separation, reinforcing the effectiveness of vMF-based diffusion in preserving hyperspherical structure.

---

**Lemma B.1** (Class Separation with von Mises-Fisher Distributions). *Consider two classes $\mathcal{C}_1$ and $\mathcal{C}_2$ on the unit hypersphere $\mathbb{S}^{d-1}$, with mean directions $\boldsymbol{\mu}_1, \boldsymbol{\mu}_2 \in \mathbb{S}^{d-1}$ separated by angle $\theta = \arccos(\boldsymbol{\mu}_1^\top \boldsymbol{\mu}_2)$. Suppose data points in each class follow von Mises-Fisher distributions with the same concentration parameter $\kappa > 0$:*

$$p(\mathbf{x}|\boldsymbol{\mu}_i, \kappa) = C_d(\kappa)\exp(\kappa\boldsymbol{\mu}_i^\top\mathbf{x}), \quad i = 1, 2 \tag{1}$$

*where, $C_d(\kappa)$ is the normalizing constant. Then:*

*(a) The probability of misclassification $P_e$ (classifying a point from class 1 as belonging to class 2 or vice versa) is bounded above by:*

$$P_e \leq \exp(-\kappa(1 - \cos\theta))$$

*(b) For any desired error rate $\epsilon > 0$, setting the concentration parameter as:*

$$\kappa \geq \frac{1}{1 - \cos\theta}\log\left(\frac{1}{\epsilon}\right)$$

*guarantees that $P_e \leq \epsilon$.*

*Proof.* 1. For a point $\mathbf{x}$ drawn from class 1, i.e., $\mathbf{x} \sim \text{vMF}(\boldsymbol{\mu}_1, \kappa)$, the probability density at angle $\phi$ from $\boldsymbol{\mu}_1$ is:

$$p(\phi) = C_d(\kappa)\exp(\kappa\cos\phi)$$

2. Misclassification occurs when $\mathbf{x}$ is closer to $\boldsymbol{\mu}_2$ than to $\boldsymbol{\mu}_1$. Byhyperspherical geometry, this happens when the angle $\phi$ between $\mathbf{x}$ and $\boldsymbol{\mu}_1$ exceeds $\theta/2$.

3. Therefore, the misclassification probability is:

$$P_e = \int_{\theta/2}^{\pi} p(\phi)\sin^{d-2}(\phi)d\phi$$

where, $\sin^{d-2}(\phi)d\phi$ is the surface element on $\mathbb{S}^{d-1}$.

4. For $\phi \geq \theta/2$, we have $\cos\phi \leq \cos(\theta/2)$, thus:

$$P_e = \int_{\theta/2}^{\pi} C_d(\kappa)\exp(\kappa\cos\phi)\sin^{d-2}(\phi)d\phi$$

$$\leq C_d(\kappa)\exp(\kappa\cos(\theta/2))\int_{\theta/2}^{\pi}\sin^{d-2}(\phi)d\phi$$

$$\leq \exp(-\kappa(1 - \cos\theta))$$

5. For part (b), solving $\exp(-\kappa(1 - \cos\theta)) \leq \epsilon$ yields the required bound on $\kappa$.

$\square$

---

***Implications:*** *(1) The bound tightens exponentially with $\kappa$. (2) Larger angular separation $\theta$ allows for smaller $\kappa$. (3) For fixed $\epsilon$, required $\kappa$ scales inversely with class separation.*

---

# C. Variational Bound

In typical hyperspherical diffusion, the reverse process attempts to invert the forward noising so that data points return precisely to their original directions. By contrast, certain tasks (e.g., class-conditional generation) may demand a less rigid constraint: as long as the final sample lies within a small *hypercone* around a class-specific prototype, the objective is satisfied. We formalize this idea by modifying the standard diffusion *variational bound* to allow for *hypercone-constrained* convergence in the reverse process.

## C.1. Forward Process on the Hypersphere

We assume data $\mathbf{z}_0 \in \mathbb{S}^{d-1}$ are sampled from some distribution $q(\mathbf{z}_0)$. The *forward noising* process gradually transforms $\mathbf{z}_0$ into an approximately uniform distribution on the hypersphere by injecting *von Mises–Fisher* noise:

$$q(\mathbf{z}_t \mid \mathbf{z}_{t-1}) = \mathrm{vMF}(\mathbf{z}_{t-1}, \kappa_t), \quad t = 1, \dots, T \tag{2}$$

where, $\mathrm{vMF}(\mu, \kappa)$ denotes a vMF distribution on $\mathbb{S}^{d-1}$ with mean direction $\mu$ and concentration parameter $\kappa \geq 0$. For large $t$, $\kappa_t \to 0$, causing the distribution $q(\mathbf{z}_T \mid \mathbf{z}_0)$ to approach *uniform* on $\mathbb{S}^{d-1}$.

Hence, the complete forward chain for $(\mathbf{z}_{0:T})$ is:

$$q(\mathbf{z}_{0:T}) = q(\mathbf{z}_0) \prod_{t=1}^{T} q(\mathbf{z}_t \mid \mathbf{z}_{t-1}) \tag{3}$$

Our goal is then to *reverse* this process, recovering $\mathbf{z}_0$ from a noisy $\mathbf{z}_T$.

## C.2. Class-Specific Hypercones and the Reverse Process

**Hypercone Definition.** We consider $C$ distinct classes, each identified by a direction $\mu_c \in \mathbb{S}^{d-1}$ and an angular radius $\theta_c \geq 0$. The class *hypercone* $\mathcal{C}_c(\theta_c)$ is then defined as:

$$\mathcal{C}_c(\theta_c) = \{ \mathbf{z} \in \mathbb{S}^{d-1} : \angle(\mathbf{z}, \mu_c) \leq \theta_c \} \tag{4}$$

Thus, each class $c$ corresponds to *all* directions on a hypersphere within angular distance $\theta_c$ of $\mu_c$.

We introduce a *class-conditional* reverse process that moves from $\mathbf{z}_T$ to $\mathbf{z}_0$ in $T$ steps:

$$p_\theta(\mathbf{z}_{0:T} \mid y) = p_\theta(\mathbf{z}_0 \mid \mathbf{z}_1, y) \prod_{t=1}^{T} p_\theta(\mathbf{z}_t \mid \mathbf{z}_{t+1}, y), \tag{5}$$

where, $y \in \{1, \dots, C\}$ is the class label. For each intermediate $t \geq 1$, we let

$$p_\theta(\mathbf{z}_{t-1} \mid \mathbf{z}_t, y) = \mathrm{vMF}(\mathbf{m}_\theta(\mathbf{z}_t, y), \tilde{\kappa}_t), \tag{6}$$

where, $\mathbf{m}_\theta(\mathbf{z}_t, y)$ is a (learned) unit vector on the hypersphere and $\tilde{\kappa}_t$ is a (possibly deterministic or learned) concentration.

**Class Hypercone at $t = 0$.** Instead of insisting that $\mathbf{z}_0$ *exactly* match the original data direction, we only require that $\mathbf{z}_0$ lie in the appropriate class hypercone $\mathcal{C}_y(\theta_y)$. Hence, we can define:

$$p_\theta(\mathbf{z}_0 \mid \mathbf{z}_1, y) = (\text{truncated vMF with support in } \mathcal{C}_y(\theta_y)), \tag{7}$$

so that $\mathbf{z}_0 \notin \mathcal{C}_y(\theta_y)$ has zero probability. Equivalently, one may define a suitable parametric distribution that is sharply peaked around $\mu_y$ but has finite support within angle $\theta_y$.

## C.3. Variational Bound with Hypercone Constraint

Let $\mathbf{z}_0$ belong to class $y$. We seek to maximize $p_\theta(\mathbf{z}_0 \mid y)$ (the likelihood of reconstructing $\mathbf{z}_0$ within the correct hypercone). A standard approach introduces the forward chain as a variational distribution and applies Jensen's inequality.

## C.4. Evidence Lower Bound (ELBO) Derivation

We start from:

$$\log p_\theta(\mathbf{z}_0 \mid y) \; = \; \log \int p_\theta(\mathbf{z}_{0:T} \mid y) \, d\mathbf{z}_{1:T} \; = \; \log \int \frac{q(\mathbf{z}_{1:T} \mid \mathbf{z}_0, y)}{q(\mathbf{z}_{1:T} \mid \mathbf{z}_0, y)} \, p_\theta(\mathbf{z}_{0:T} \mid y) \, d\mathbf{z}_{1:T}, \tag{8}$$

where, $q(\mathbf{z}_{1:T} \mid \mathbf{z}_0, y) = \prod_{t=1}^{T} q(\mathbf{z}_t \mid \mathbf{z}_{t-1}, y)$, but note that in practice $q(\mathbf{z}_t \mid \mathbf{z}_{t-1}, y)$ usually coincides with $q(\mathbf{z}_t \mid \mathbf{z}_{t-1})$ if the forward noising is *class-agnostic*. Applying Jensen's inequality to the expression inside :

$$\log p_\theta(\mathbf{z}_0 \mid y) = \log \mathbb{E}_{q(\mathbf{z}_{1:T} \mid \mathbf{z}_0, y)} \left[ \frac{p_\theta(\mathbf{z}_{0:T} \mid y)}{q(\mathbf{z}_{1:T} \mid \mathbf{z}_0, y)} \right] \; \geq \; \mathbb{E}_{q(\mathbf{z}_{1:T} \mid \mathbf{z}_0, y)} \left[ \log \frac{p_\theta(\mathbf{z}_{0:T} \mid y)}{q(\mathbf{z}_{1:T} \mid \mathbf{z}_0, y)} \right] \tag{9}$$

Hence we define the negative ELBO:

$$\mathcal{L}(\theta) \; = \; \mathbb{E}_{q(\mathbf{z}_{0:T} \mid \mathbf{z}_0, y)} \left[ \log \frac{q(\mathbf{z}_{1:T} \mid \mathbf{z}_0, y)}{p_\theta(\mathbf{z}_{0:T} \mid y)} \right], \quad \text{so that} \quad \log p_\theta(\mathbf{z}_0 \mid y) \; \geq \; -\mathcal{L}(\theta) \tag{10}$$

## C.5. Decomposition

By writing out $q(\mathbf{z}_{1:T} \mid \mathbf{z}_0, y)$ and $p_\theta(\mathbf{z}_{0:T} \mid y)$ explicitly, we get:

$$q(\mathbf{z}_{1:T} \mid \mathbf{z}_0, y) = \prod_{t=1}^{T} q(\mathbf{z}_t \mid \mathbf{z}_{t-1}, y),$$

$$p_\theta(\mathbf{z}_{0:T} \mid y) = p_\theta(\mathbf{z}_0 \mid \mathbf{z}_1, y) \prod_{t=1}^{T} p_\theta(\mathbf{z}_t \mid \mathbf{z}_{t+1}, y)$$

Therefore,

$$\log \frac{q(\mathbf{z}_{1:T} \mid \mathbf{z}_0, y)}{p_\theta(\mathbf{z}_{0:T} \mid y)} \; = \; -\log p_\theta(\mathbf{z}_0 \mid \mathbf{z}_1, y) \; + \; \sum_{t=1}^{T} \Big[ \log q(\mathbf{z}_t \mid \mathbf{z}_{t-1}, y) \; - \; \log p_\theta(\mathbf{z}_t \mid \mathbf{z}_{t+1}, y) \Big] \tag{11}$$

Taking expectation under $q(\mathbf{z}_{0:T} \mid \mathbf{z}_0, y)$ yields:

$$\mathcal{L}(\theta) \; = \; \underbrace{\mathbb{E}_{q(\mathbf{z}_{0:1} \mid \mathbf{z}_0, y)} \big[ -\log p_\theta(\mathbf{z}_0 \mid \mathbf{z}_1, y) \big]}_{\text{(reconstruction into hypercone)}} + \sum_{t=1}^{T} \underbrace{\mathbb{E}_{q(\mathbf{z}_{0:T} \mid \mathbf{z}_0, y)} \Big[ \log q(\mathbf{z}_t \mid \mathbf{z}_{t-1}, y) \; - \; \log p_\theta(\mathbf{z}_t \mid \mathbf{z}_{t+1}, y) \Big]}_{\text{(KL terms between forward vMF and reverse vMF)}} \tag{12}$$

**Hypercone Constraint.** The key difference from standard hyperspherical diffusion is that

$$p_\theta(\mathbf{z}_0 \mid \mathbf{z}_1, y) \text{ is constrained to } \mathcal{C}_y(\theta_y),$$

i.e. we ensure that $\mathbf{z}_0$ stays within angular distance $\theta_y$ of the class mean $\mu_y$. Hence the "reconstruction term" $-\log p_\theta(\mathbf{z}_0 \mid \mathbf{z}_1, y)$ does not drive the model to a *single point* $\mu_y$, but rather to the full hypercone $\mathcal{C}_y(\theta_y)$. Mathematically, this can be implemented by a truncated vMF distribution or any parametric distribution that has *zero probability* outside $\angle(\mathbf{z}, \mu_y) > \theta_y$.

## C.6. Resulting Objective

Combining the above, we arrive at:

$$\log p_\theta(\mathbf{z}_0 \mid y) \; \geq \; -\mathcal{L}(\theta) \; = \; -\mathbb{E}_{q(\mathbf{z}_{1:T} \mid \mathbf{z}_0, y)} \Big[ \log p_\theta(\mathbf{z}_0 \mid \mathbf{z}_1, y) \; - \; \sum_{t=1}^{T} \Big( \log p_\theta(\mathbf{z}_t \mid \mathbf{z}_{t+1}, y) \; - \; \log q(\mathbf{z}_t \mid \mathbf{z}_{t-1}, y) \Big) \Big] \tag{13}$$

If each $p_\theta(\mathbf{z}_t \mid \mathbf{z}_{t+1}, y)$ is a $\text{vMF}(\mathbf{m}_\theta, \tilde{\kappa}_t)$ and each $q(\mathbf{z}_t \mid \mathbf{z}_{t-1}, y)$ is $\text{vMF}(\mathbf{z}_{t-1}, \kappa_t)$, then each KL term has a known closed form. The final $t = 0$ term effectively enforces that $\mathbf{z}_0$ remains in the class hypercone. Thus, the chain *converges to a distribution* localized around $\mu_y$, with an angular radius $\theta_y$, rather than collapsing to a single direction.

# D. Uncertainty Modelling

**Lemma D.1** (Equivalence of Angular Interpolation and vMF Diffusion). *Let $\mathbf{z}_t \in \mathbb{S}^{d-1}$ be generated by either:*

1. *Angular interpolation: $\mathbf{z}_t = \cos(\theta_t)\mathbf{z}_{t-1} + \sin(\theta_t)\mathbf{v}$, where $\mathbf{v} \sim Uniform(\mathbb{S}^{d-1})$*

2. *vMF sampling: $\mathbf{z}_t \sim vMF(\mathbf{z}_{t-1}, \kappa_t)$*

*For $\kappa_t = \cot(\theta_t)$, these processes generate equivalent distributions over the hypersphere.*

*Proof.* Under angular interpolation, $\theta_t = 0$ gives $\mathbf{z}_t = \mathbf{z}_{t-1}$ (perfect preservation), while $\theta_t = \pi/2$ gives $\mathbf{z}_t = \mathbf{v}$ (uniform noise). For vMF with $\kappa_t = \cot(\theta_t)$, these correspond to $\kappa_t \to \infty$ (perfect concentration) and $\kappa_t \to 0$ (uniform distribution) respectively. The equivalence follows from the conditional density:

$$p(\mathbf{z}_t|\mathbf{z}_{t-1}) \propto \exp(\cot(\theta_t)\mathbf{z}_{t-1}^\top \mathbf{z}_t)$$

which matches the vMF density $f(\mathbf{z}_t; \mathbf{z}_{t-1}, \kappa_t) \propto \exp(\kappa_t \mathbf{z}_{t-1}^\top \mathbf{z}_t)$ when $\kappa_t = \cot(\theta_t)$. $\qquad\square$

*Implications:* *This equivalence offers geometric (angular interpolation) and probabilistic (vMF) views of the forward process, with $\kappa_t = \cot(\theta_t)$ ensuring compatibility with Smooth progression $\theta_t : 0 \to \pi/2$ matches $\kappa_t : \infty \to 0$*

---

**Lemma D.2** (**Concentration of vMF Reverse Process into a Class Hypercone**). *Suppose we have a discrete-time reverse Markov chain $\{\mathbf{z}_t\}_{t=T}^0 \subset \mathbb{S}^{d-1}$ defined by*

$$\mathbf{z}_{t-1} \sim \mathrm{vMF}\Big( \Pi\big(\mathbf{z}_t + \eta_t \nabla_{\mathbf{z}_t} \log f(\mathbf{z}_t; \boldsymbol{\mu}_c)\big), \kappa_t \Big) \tag{14}$$

*where:*

1. *$\Pi(\mathbf{x}) := \mathbf{x}/\|\mathbf{x}\|$ is the projection onto the unit hypersphere $\mathbb{S}^{d-1}$.*

2. *$\nabla_{\mathbf{z}_t} \log f(\mathbf{z}_t; \boldsymbol{\mu}_c)$ is the gradient (score function) of a density $f(\mathbf{z}; \boldsymbol{\mu}_c)$ that is sharply peaked around the class mean $\boldsymbol{\mu}_c \in \mathbb{S}^{d-1}$. In particular, this gradient points largely in the direction of $\boldsymbol{\mu}_c - \mathbf{z}_t$ (where $\mathbf{u} = \boldsymbol{\mu}_c - \mathbf{z}_t$ ) whenever $\mathbf{z}_t$ is not too close to $\boldsymbol{\mu}_c$.*

3. *$\kappa_t$ (the vMF concentration) increases over time, and $\eta_t \to 0$ at a suitable rate (e.g. $\eta_t \kappa_t \to \infty$ but $\eta_t \to 0$ as $t \to 0$).*

*Define the* class hypercone *of half-angle $\alpha > 0$ around $\boldsymbol{\mu}_c$ by:*

$$\mathcal{C}_\alpha(\boldsymbol{\mu}_c) := \Big\{ \mathbf{u} \in \mathbb{S}^{d-1} \ \Big| \ \mathbf{u}^\top \boldsymbol{\mu}_c \geq \cos(\alpha) \Big\}$$

*Then under mild smoothness assumptions on $f$ and the above monotonicity/decay rates for $\kappa_t$ and $\eta_t$, the chain converges (in distribution) into the hypercone $\mathcal{C}_\alpha(\boldsymbol{\mu}_c)$ as $t \to 0$. Specifically, for any $\alpha > 0$,*

$$\lim_{t \to 0} \mathbb{P}\big[ \mathbf{z}_t \in \mathcal{C}_\alpha(\boldsymbol{\mu}_c) \big] = 1$$

*Equivalently, the angle between $\mathbf{z}_t$ and $\boldsymbol{\mu}_c$ goes to zero with high probability.*

*Sketch of Proof.* We outline the main arguments:

1. **Directional Gradient Alignment.** By assumption, $\nabla_{\mathbf{z}_t} \log f(\mathbf{z}_t; \boldsymbol{\mu}_c)$ points generally toward $\boldsymbol{\mu}_c$ for $\mathbf{z}_t$ not close to $\boldsymbol{\mu}_c$. Hence the update $\mathbf{z}_t + \eta_t \nabla_{\mathbf{z}_t} \log f$ rotates $\mathbf{z}_t$ closer in angle to $\boldsymbol{\mu}_c$. After projection $\Pi(\cdot)$ to the hypersphere, this remains a unit vector closer to $\boldsymbol{\mu}_c$ than $\mathbf{z}_t$ was.

2. **Sharp Concentration under vMF.** As $\kappa_t \to \infty$, $\mathrm{vMF}(\mathbf{m}, \kappa_t)$ puts most of its mass around $\mathbf{m}$, with the variance of the angular distribution shrinking like $1/\kappa_t$. Thus, if the mean direction $\Pi(\mathbf{z}_t + \eta_t \nabla_{\mathbf{z}_t} \log f)$ is already within $\alpha$ of $\boldsymbol{\mu}_c$, then with high probability the new sample $\mathbf{z}_{t-1}$ remains near $\boldsymbol{\mu}_c$.

3. **Iteration and Convergence.** Given that $\eta_t \to 0$, the movement per step in the hypersphere's tangent space shrinks, preventing large excursions away from $\boldsymbol{\mu}_c$. Meanwhile, $\kappa_t$ (the concentration) grows so that each step's sample is drawn from an increasingly peaked distribution. Iterating backward from $t = T$ down to $t = 0$, the probability of $\mathbf{z}_t$ being outside any cone $\mathcal{C}_\alpha(\boldsymbol{\mu}_c)$ diminishes at each step. By the last steps near $t = 0$, $\mathbf{z}_t$ concentrates with high probability in the chosen hypercone around $\boldsymbol{\mu}_c$.

Thus, in the limit $t \to 0$, we conclude that $\mathbf{z}_t$ converges in distribution to directions arbitrarily close to $\boldsymbol{\mu}_c$, i.e. within any desired half-angle $\alpha$. Equivalently, the angle $\angle(\mathbf{z}_t, \boldsymbol{\mu}_c)$ goes to zero with high probability, implying $\lim_{t \to 0} \mathbf{z}_t^\top \boldsymbol{\mu}_c = 1$ almost surely. $\qquad\square$

---

**Algorithm 3** Hypercone-Constrained Sampling with Learned Truncation

---

**Require:** Class label $y$, diffusion steps $T$
**Require:** Angular schedule $\{\theta_t\}_{t=1}^T$, step sizes $\{\eta_t\}_{t=1}^T$
 1: Sample $\mathbf{z}_T \sim \text{Uniform}(\mathbb{S}^{d-1})$
 2: **for** $t = T$ to $1$ **do**
 3: $\quad \mathbf{m}_t \leftarrow D_\phi(\mathbf{z}_t, t, y)$ $\qquad\qquad\qquad\qquad\qquad\qquad\qquad\qquad\qquad$ ▷ Predict direction
 4: $\quad \theta_y \leftarrow C_\psi(\mathbf{z}_t, t, y)$ $\qquad\qquad\qquad\qquad\qquad\qquad\qquad\qquad\qquad$ ▷ Predict angle
 5: $\quad \mathcal{C}_{t,y} \leftarrow \{\mathbf{z} : \angle(\mathbf{z}, \mathbf{m}_t) \leq \theta_y\}$
 6: $\quad \phi_t \leftarrow \angle(\mathbf{z}_t, \mathbf{m}_t)$
 7: $\quad \kappa_t \leftarrow \kappa_{\max}\sigma(\beta[\theta_y - \phi_t])$
 8: $\quad$ **if** $t > 1$ **then**
 9: $\quad\quad \nabla_{\mathbf{z}_t} \log f \leftarrow \text{ScoreNet}_\theta(\mathbf{z}_t, t, y)$
10: $\quad\quad \mathbf{u}_t \leftarrow \mathbf{z}_t + \eta_t \nabla_{\mathbf{z}_t} \log f$
11: $\quad\quad \mathbf{u}_t \leftarrow \mathbf{u}_t / \|\mathbf{u}_t\|$
12: $\quad\quad \mathbf{z}_{t-1} \sim \text{TvMF}(\mathbf{u}_t, \kappa_t, \mathcal{C}_{t,y})$
13: $\quad$ **end if**
14: **end for**
15: **return** Final sample $\mathbf{z}_1$

---

# E. Brownian Motion on Hypersphere

Recent advances in continuous-time score-based generative models (Song et al., 2021; Karras et al., 2022) suggest viewing the forward noising process as an *SDE*, whose time-reversal recovers the data distribution. When data reside on a hypersphere $\mathbb{S}^{d-1}$, an analogous approach involves constructing a *Brownian motion* restricted to $\mathbb{S}^{d-1}$, then reversing it to produce samples from the original (or a conditional) distribution. Concretely, let $\mathbf{z}(t) \in \mathbb{S}^{d-1}$ evolve for $t \in [0, T]$ such that it follows an *intrinsic* Brownian motion on the hypersphere. In the simplest form,

$$d\mathbf{z}(t) = \sqrt{2\,\sigma^2(t)}\,\mathbf{P}_{\mathbf{z}(t)}\,d\mathbf{w}(t), \tag{15}$$

where, $\sigma(t) \geq 0$ is a noise scale (or diffusion coefficient) and $\mathbf{P}_{\mathbf{z}(t)}$ projects $\mathbb{R}^d$ increments $d\mathbf{w}(t)$ onto the tangent space at $\mathbf{z}(t) \in \mathbb{S}^{d-1}$.

**Spherical Forward Process and Hyperspherical Score Estimation.** By choosing $\sigma(t)$ such that the marginal distribution of $\mathbf{z}(T)$ approaches the uniform measure on $\mathbb{S}^{d-1}$, we obtain a continuous analog of spherical diffusion. In practice, one can incorporate a small drift term $b(\mathbf{z}, t)$ to ensure that $q(\mathbf{z}(T))$ is nearly uniform, similar to the variance-preserving or variance-exploding schedules in Euclidean SDEs (Song et al., 2021). Concurrently, we learn a *score network* $s_\theta(\mathbf{z}, t)$ that approximates $\nabla_{\mathbf{z}} \log q_t(\mathbf{z})$, using a spherical variant of score matching (Vincent, 2011).

**Reverse-Time SDE and Generative Sampling.** Once $s_\theta$ is trained, we generate new samples by solving the reverse-time SDE. Formally, time reversal of the forward process (15) yields

$$d\bar{\mathbf{z}}(t) = \left[-\sigma^2(t)\,s_\theta(\bar{\mathbf{z}}, t)\right] dt + \sqrt{2\,\sigma^2(t)}\,\mathbf{P}_{\bar{\mathbf{z}}(t)}\,d\bar{\mathbf{w}}(t),$$

where, $t \in [T, 0]$ and $d\overline{\mathbf{w}}(t)$ is again Brownian noise on the tangent space, and we solve backward from $\overline{\mathbf{z}}(T) \sim$ Uniform($\mathbb{S}^{d-1}$) down to $\overline{\mathbf{z}}(0)$. Intuitively, the term $-\sigma^2(t) s_\theta(\overline{\mathbf{z}}, t)$ acts as a drift that guides samples toward the data manifold on the hypersphere.

**Comparison to vMF Diffusion.** In discrete-time spherical diffusion using vMF noise , each forward step is $\text{vMF}(\mathbf{z}_{t-1}, \kappa_t)$, while the reverse step estimates $\text{vMF}(\cdot, \tilde{\kappa}_t)$ with a learned center. Spherical Brownian motion (15) can be viewed as the *continuous limit* of many small vMF perturbations. Inversely, discretizing (E) via Euler–Maruyama method yields a small-angle vMF reverse step that remains tangent to $\mathbb{S}^{d-1}$.

**Lemma E.1.** ***Discrete-Continuous Correspondence*** *Let $q_{\Delta t}(\mathbf{z}_{t+\Delta t}|\mathbf{z}_t)$ be a vMF transition with concentration $\kappa \Delta t$. As $\Delta t \to 0$ with $\kappa_{\Delta t} = O(1/\Delta t)$, the process converges weakly to the solution of the spherical Brownian motion SDE (15).*

**Handling Class Hypercones and Adaptive Freezing.** Finally, to respect class geometry or hypercone constraints, one may introduce a drift term that becomes very large (or a reflection boundary) whenever $\mathbf{z}(t)$ moves outside the class-constrained cone. Alternatively, if each class $y$ has a known center $\boldsymbol{\mu}_y$ and angular tolerance $\theta_y$, the reverse SDE (E) can incorporate a penalty encouraging $\angle(\mathbf{z}, \boldsymbol{\mu}_y) \le \theta_y$, effectively "locking" ( constraining to a sub-manifold like a hypercone on the sphere) trajectories in the appropriate region of $\mathbb{S}^{d-1}$. Such mechanisms ensure the final sample remains class-consistent while leveraging the flexibility and elegance of spherical Brownian motion as the underlying diffusion.

## F. Beyond Geometry and Uncertainty: Retaining Image Information

For the experiment with two diffusion process, the **forward** (noising) step at time $t$ can be viewed as:

$$\alpha_t = \alpha_{t-1} + \sigma\,\epsilon_t; \quad \mathbf{d}_t \sim \text{vMF}(\mathbf{d}_{t-1}, \kappa_t)$$
$$\mathbf{x}_t = \alpha_t\,\mathbf{d}_t$$

where, $\sigma\,\epsilon_t$ is a small Gaussian perturbation (e.g. $\epsilon_t \sim \mathcal{N}(0,1)$) controlling how the magnitude spreads, and $\mathbf{d}_t$ is sampled from a vMF distribution centered at $\mathbf{d}_{t-1}$ with concentration $\kappa_t$. As $t$ increases, $\kappa_t$ may decrease (making directions more diffuse), while $\sigma$ can enlarge the variance of $\alpha_t$, allowing the embedding norm to fluctuate more widely.

**Reverse (Denoising) Process.** We define the reverse chain to invert both magnitude and direction back to the original class-consistent configuration. To invert this process, we learn parameters $\theta$ that predict a suitable Gaussian mean for the magnitude and a suitable mean direction for the vMF. Formally, we define

$$\widehat{\alpha}_{t-1} \sim \mathcal{N}\Big(\mu_\theta(\alpha_t, \mathbf{d}_t), \widetilde{\sigma}^2\Big),$$

$$\widehat{\mathbf{d}}_{t-1} \sim \text{vMF}\Big(\mathbf{m}_\theta(\alpha_t, \mathbf{d}_t), \widetilde{\kappa}_t\Big),$$

$$\mathbf{x}_{t-1} = \widehat{\alpha}_{t-1}\,\widehat{\mathbf{d}}_{t-1}$$

Here, $\mu_\theta(\cdot)$ is a neural network output that infers the ideal norm given the current $\alpha_t$ and $\mathbf{d}_t$, while $\mathbf{m}_\theta(\cdot) \in \mathbb{S}^{d-1}$ is another network output that estimates the best directional center for the vMF.

This design provides a powerful way to incorporate both **global image information** and **class geometry**: we learn how to *denoise* both magnitude (via the Gaussian) and direction (via vMF), thus recovering the class-relevant structure on the hypersphere encoded in the angular direction while preserving essential image information encoded in $\alpha$. This dual-uncertainty approach ensures that face embeddings can vary naturally in intensity or brightness, yet remain consistent with class geometry, capturing both *how bright* an image is and *which identity* it belongs to.

---

**Theorem F.1.** ***Dual-Uncertainty Preservation*** *Let $\mathbf{x}_0 \in \mathbb{R}^d$ with $\alpha_0 = ||\mathbf{x}_0||$ and $\mathbf{d}_0 = \mathbf{x}_0/||\mathbf{x}0||$. Under the forward process:*

$$\alpha_t = \alpha_{t-1} + \sigma\epsilon_t,$$
$$\epsilon_t \sim \mathcal{N}(0,1) \; \mathbf{d}t \sim \text{vMF}(\mathbf{d}_{t-1}, \kappa_t) \; \mathbf{x}_t = \alpha_t\mathbf{d}t$$

*The following holds for all $t \geq 0$:*

$$\mathbb{E}[\alpha_t - \alpha_0] = 0$$

$$\mathrm{Var}(\alpha_t - \alpha_0) = t\sigma^2 \mathbb{E}[\mathbf{d}t^\top \mathbf{d}0] = \prod_{i=1}^{t} A_d(\kappa_i)$$

*where, $A_d(\kappa) = I_{d/2}(\kappa)/I_{d/2-1}(\kappa)$ is the ratio of modified Bessel functions. Moreover, $p(\mathbf{x}_t|\mathbf{x}_0) = p(\alpha_t|\alpha_0)p(\mathbf{d}_t|\mathbf{d}_0)$ with $\mathbf{d}_t \in \mathbb{S}^{d-1}$ almost surely.*

---

## G. Adaptive Class-based Hypercone Learning

**Class Hypercone Setup.** Each class $y \in \{1, \ldots, C\}$ is associated with a direction $\boldsymbol{\mu}_y \in \mathbb{S}^{d-1}$ and an angular radius $\theta_y \geq 0$. Hence, the class hypercone is defined by:

$$\mathcal{C}_y(\theta_y) = \left\{ \mathbf{z} \in \mathbb{S}^{d-1} : \angle(\mathbf{z}, \boldsymbol{\mu}_y) \leq \theta_y \right\} \tag{16}$$

**Forward (Noising) Process.** We consider a forward chain of length $T$, where each step injects vMF noise:

$$q(\mathbf{z}_{1:T} \mid \mathbf{z}_0) = \prod_{t=1}^{T} q(\mathbf{z}_t \mid \mathbf{z}_{t-1}), \quad \text{where} \quad q(\mathbf{z}_t \mid \mathbf{z}_{t-1}) = \mathrm{vMF}(\mathbf{z}_{t-1}, \kappa_t) \tag{17}$$

Here, $\kappa_t$ is a (potentially decreasing) schedule that pushes $\mathbf{z}_t$ toward uniform on $\mathbb{S}^{d-1}$ as $t$ grows.

**Adaptive Reverse with Learned Concentration.** Instead of using a fixed reverse schedule, we let

$$\kappa_\theta(\mathbf{z}_t, y) : \mathbb{S}^{d-1} \times \{1, \ldots, C\} \to \mathbb{R}_{\geq 0} \tag{18}$$

be a learned function of the current state $\mathbf{z}_t \in \mathbb{S}^{d-1}$ and the class $y$. We define the reverse model as

$$p_\theta(\mathbf{z}_{0:T} \mid y) = p(\mathbf{z}_T) \prod_{t=1}^{T} p_\theta(\mathbf{z}_{t-1} \mid \mathbf{z}_t, y), \text{where} \quad p_\theta(\mathbf{z}_{t-1} \mid \mathbf{z}_t, y) = \mathrm{vMF}\Big(\mathbf{m}_\theta(\mathbf{z}_t, y), \kappa_\theta(\mathbf{z}_t, y)\Big) \tag{19}$$

Here, $p(\mathbf{z}_T)$ is uniform on the hypersphere (i.e. the limiting vMF with zero concentration), and $\mathbf{m}_\theta(\mathbf{z}_t, y) \in \mathbb{S}^{d-1}$ is a learned mean direction. The key distinction is that $\kappa_\theta(\mathbf{z}_t, y)$ can grow large once $\mathbf{z}_t$ is in the correct hypercone, effectively "freezing" further denoising.

**Example of a Freezing Mechanism.** Let

$$\alpha(\mathbf{z}_t, y) = \angle(\mathbf{z}_t, \boldsymbol{\mu}_y) \text{and} \quad \kappa_\theta(\mathbf{z}_t, y) = \kappa_{\max} \sigma\Big(\beta\left[\theta_y - \alpha(\mathbf{z}_t, y)\right]\Big), \tag{20}$$

where, $\sigma(\cdot)$ is a monotonic squashing function (e.g. sigmoid), $\kappa_{\max}$ is a large positive constant, and $\beta > 0$ controls slope. If $\alpha(\mathbf{z}_t, y) \leq \theta_y$, then $\kappa_\theta(\mathbf{z}_t, y)$ saturates near $\kappa_{\max}$, locking $\mathbf{z}_{t-1}$ into the hypercone $\mathcal{C}_y(\theta_y)$ by making the vMF distribution highly concentrated.

## H. Experimental Details

For feature extraction, we employ different architectures based on the domain: facial representations are obtained using a pre-trained ArcFace (Deng et al., 2019) model with iResNet50 (Duta et al., 2021) backbone, while object categories use a CNN-based feature extractor, both producing hyperspherical embeddings. The latent space is configured with dimensions $32 \times 32 \times 3$, balancing detail and computational efficiency. Our diffusion process uses an angular parameter $\theta$ that progresses from 0 to $\pi/2$ across diffusion steps, with the concentration parameter $\kappa$ derived as $\cot(\theta)$, naturally transitioning from high concentration ($\kappa >> 1$ at $\theta = 0$) to a uniform distribution. For class-conditional generation, we utilize a context dimension of 512 to encode class information. The class specific hypercone constraints are implemented through adaptive $\kappa$

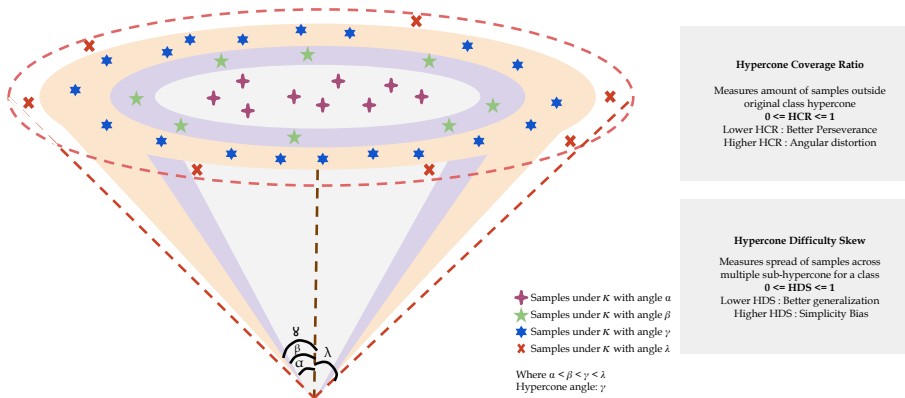

*Figure 8.* Geometric interpretation of proposed metrics HCR and HDS.

values, ranging from 0 to a class-specific maximum determined by the class angular radius $\theta_c$, which is predicted by a UNet architecture. The maximum $\kappa$ follows $\kappa_{max} = \log(1/\epsilon)/(1 - \cos(\theta_c))$, ensuring generated samples respect class-specific geometric structure. For training, we use the Adam optimizer with a learning rate of 1e-4 and batch size of 128, with the model trained for 100K iterations on a single NVIDIA A100 GPU. For comparison with Gaussian based diffusion, we used Variance preserving variant of diffusion.

### H.1. Analysis of metrics

The Hypercone Coverage Ratio (HCR) and Hypercone Difficulty Skew (HDS) together offer a comprehensive framework for evaluating generative models in terms of their alignment with the original class distribution and their ability to handle sample difficulty.

The HCR primarily assesses whether the generative model preserves the class structure by examining the percentage of generated samples that fall outside the class's hypercone. A lower HCR suggests that the model is generating samples that remain within the expected angular region of the class distribution, indicating that it faithfully reproduces the structure of the original class. On the other hand, a higher HCR implies that the model is producing out-of-class or unrealistic samples that deviate significantly from the class's expected region. This could be a sign of overfitting or a failure to learn the true distribution of the class, leading to unrealistic or poorly sampled outputs.

The HDS, in contrast, focuses on the model's ability to balance difficulty across the generated samples. By dividing the class's hypercone into multiple sub-cones based on increasing angular thresholds, HDS captures how well the model distributes its generated samples across regions of varying difficulty. The model is expected to generate a mix of easy (close to the class mean) and hard (further from the mean) samples, reflecting the full diversity of the class. A high HDS indicates that the model primarily generates easy samples clustered in the innermost sub-cones. This could suggest a bias towards overfitting simpler patterns. This bias is undesirable because it fails to capture the full range of complexity in the class distribution. Conversely, a low HDS suggests that the model is distributing its samples more evenly across different difficulty levels, which is indicative of a more robust generative process that captures both simple and complex variations in the class.

Together, these two metrics help to paint a fuller picture of a generative model's performance. A well-balanced generative model should ideally have a low HCR, reflecting good preservation of class boundaries, and a moderate HDS, indicating that it generates a variety of sample difficulties, capturing the full complexity of the class distribution. A high HCR combined with a low HDS might indicate that the model is overly focused on easy samples, while a high HDS with a low HCR could suggest that the model is struggling to maintain class consistency. Thus, an optimal model should strike a balance, maintaining a good coverage of the class's angular space without overfitting to simpler, easier samples.

### H.2. Results

**Reverse Denoising Comparison** The results presented in Tables 3 to 5 provide a comparative analysis of Euclidean versus angular-based reverse denoising strategies across three datasets: CIFAR-10, MNIST, and D-LORD. For all datasets, we observe that angular-based methods, particularly those using cosine and geodesic formulations, consistently offer

improvements or comparable performance across key metrics. Specifically, angular approaches yield lower FID scores, indicating better sample quality. For example, FID improves from 3.52 to 3.28 on CIFAR-10 and from 1.86 to 1.77 on MNIST. In terms of Hypercone Coverage Ratio (HCR), angular losses preserve class structure similarly or better than MSE-based approaches, while Hypercone Difficulty Skew (HDS) values suggest that angular methods produce a more balanced range of sample difficulty. These improvements highlight that angular loss functions better align with the intrinsic hyperspherical nature of feature embeddings, enhancing both generation quality and class consistency. The consistency of these trends across datasets confirms the generalizability of angular reverse denoising in hyperspherical diffusion models.

*Table 3.* Comparative analysis of Euclidean and Angular based reverse denoising step for CIFAR-10 dataset. Various score functions are also used for evaluation.

|     | Euclidean with MSE | Angular with Cosine | Angular with Geodesic |
|-----|--------------------|---------------------|-----------------------|
| FID | 3.52               | 3.28                | 3.35                  |
| HCR | 0.20               | 0.20                | 0.19                  |
| HDS | 0.48               | 0.51                | 0.47                  |

*Table 4.* Comparative analysis of Euclidean and Angular based reverse denoising step for MNIST dataset.

|     | Euclidean with MSE | Angular with Cosine | Angular with Geodesic |
|-----|--------------------|---------------------|-----------------------|
| FID | 1.86               | 1.79                | 1.77                  |
| HCR | 0.14               | 0.15                | 0.14                  |
| HDS | 0.52               | 0.47                | 0.48                  |

*Table 5.* Comparative analysis of Euclidean and Angular based reverse denoising step for D-LORD dataset.

|     | Euclidean with MSE | Angular with Cosine | Angular with Geodesic |
|-----|--------------------|---------------------|-----------------------|
| FID | 9.27               | 9.01                | 8.97                  |
| HCR | 0.21               | 0.19                | 0.20                  |
| HDS | 0.62               | 0.61                | 0.62                  |

**Effect of Scheduling $\kappa$** Scheduling $\kappa$ controls the rate at which class structure degrades, ensuring a smooth transition to uniform noise. Without scheduling, any fixed $\kappa$ eventually results in a uniform distribution on the hypersphere, particularly as $T$ increases. Gradually decaying $\kappa_t$ preserves intra-class structure longer, aiding recovery during the reverse process. Formally, for $\mathbf{d}t \sim \text{vMF}(\mathbf{d}t - 1, \kappa_t)$, the marginal distribution approaches uniformity as $\kappa_T \to 0 \quad p(\mathbf{d}_T) \approx \frac{1}{|\mathbb{S}^{d-1}|}$. Empirical results on the effect of $\kappa$ scheduling are shown in Table 6.

**Feature Representation** Figure 9 illustrates the feature representations of conditional samples generated from the 10 classes of the CIFAR-10 dataset. The figure highlights that the vMF-based reverse sampling effectively converges samples within class-specific hypercones, capturing the angular geometry of the data. In contrast, Gaussian-based reverse sampling produces samples that converge within Euclidean space, failing to adhere to the hyperspherical structure.

**Facial Data Generation** The Figure 10 illustrates the generated images with diversity and occlusion facial challenges present, which is critical for robust face recognition under real-world conditions. The top section presents multiple views of the same subject generated without occlusion, showing typical intra-subject variation. The middle section focuses on cases with eye-region occlusion caused by sunglasses, while the bottom section includes examples of full occlusion from multiple accessories such as hats and scarves.

**Hypercone specific generation** Figure 12 and Figure 13 shows samples generated from inner and outer hypercone based on class-specific $\theta_k$ that determines the boundary of class. As demonstrated by the figure, the images sampled from inner hypercone are sharp and realistic while samples generated around the boundary are noisy. Also, various samples generated are shown in Figure 11

## I. Applications

The applications of the proposed vMF-based angular diffusion are:

*Table 6.* Effect of using kappa scheduler. Comparative analysis of using the kappa scheduler on two datasets for various evaluation metrics

| Metric | CIFAR-10 | | MNIST | |
|---|---|---|---|---|
| | With scheduling | Without scheduling | With scheduling | Without scheduling |
| Class-wise Accuracy | 89.35 | 72.59 | 96.01 | 86.48 |
| FID | 3.52 | 6.28 | 1.86 | 2.11 |
| HCR | 0.20 | 0.37 | 0.14 | 0.21 |
| HDS | 0.48 | 0.63 | 0.52 | 0.75 |

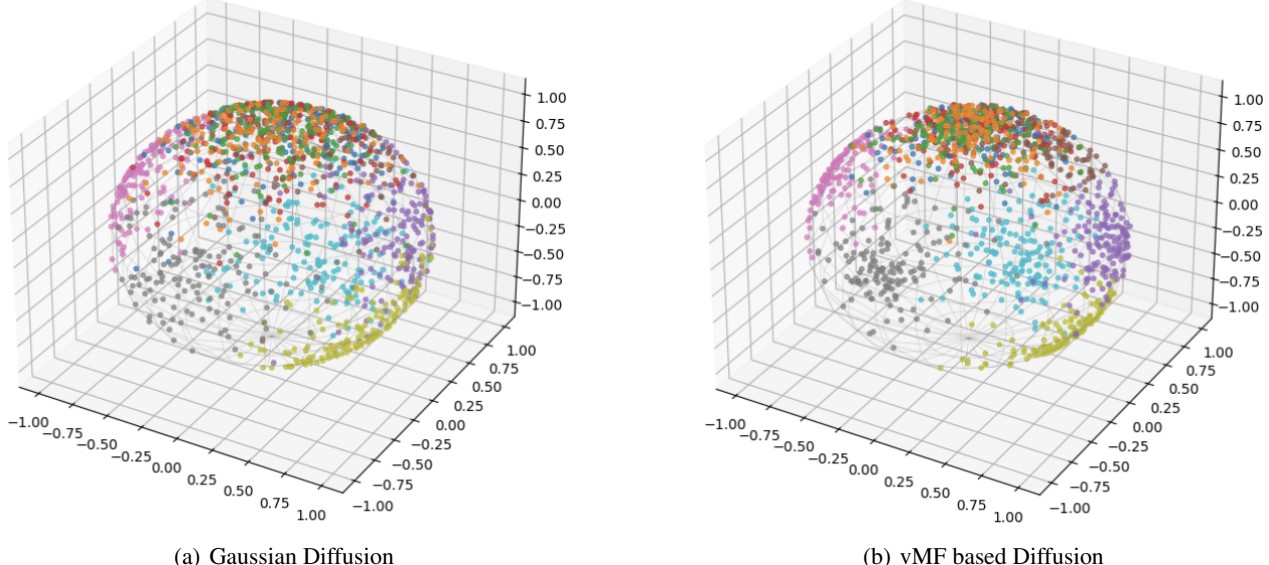

(a) Gaussian Diffusion        (b) vMF based Diffusion

*Figure 9.* Feature representation of the CIFAR-10 dataset generated using Gaussian-based diffusion (left) and vMF-based diffusion (right). The vMF-based sampling aligns generated sample features within class-specific 3D hypercones, while Gaussian-based sampling results in scattered features outside the class-hypercones.

- Few-shot learning: Our approach improves performance by generating more diverse and class-consistent samples from limited data.

- Fairness and bias mitigation: Manifold-aware generation allows controlled augmentation to rebalance datasets across demographics, reducing biases.

- Face recognition robustness: Explicitly preserving directional structures helps models robustly handle variations (occlusion, illumination, pose).

- Difficult sample generation: Controlled angular diffusion produces challenging samples near class boundaries, refining decision boundaries and improving model reliability.

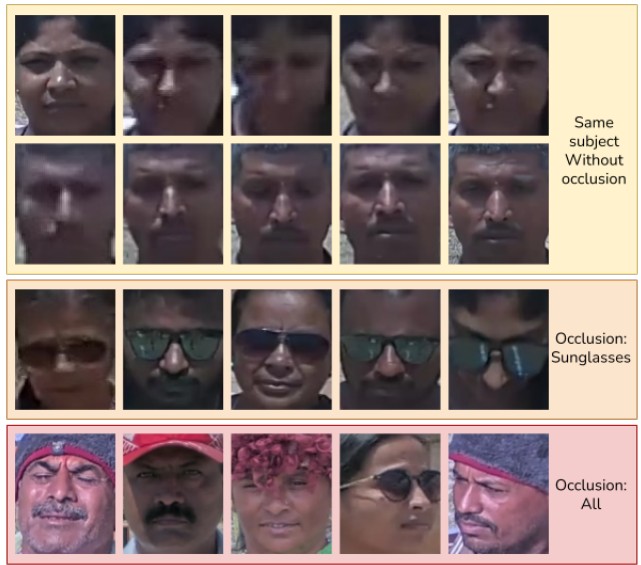

*Figure 10.* Facial data synthesis demonstrates variations across occlusion, pose and resolution.

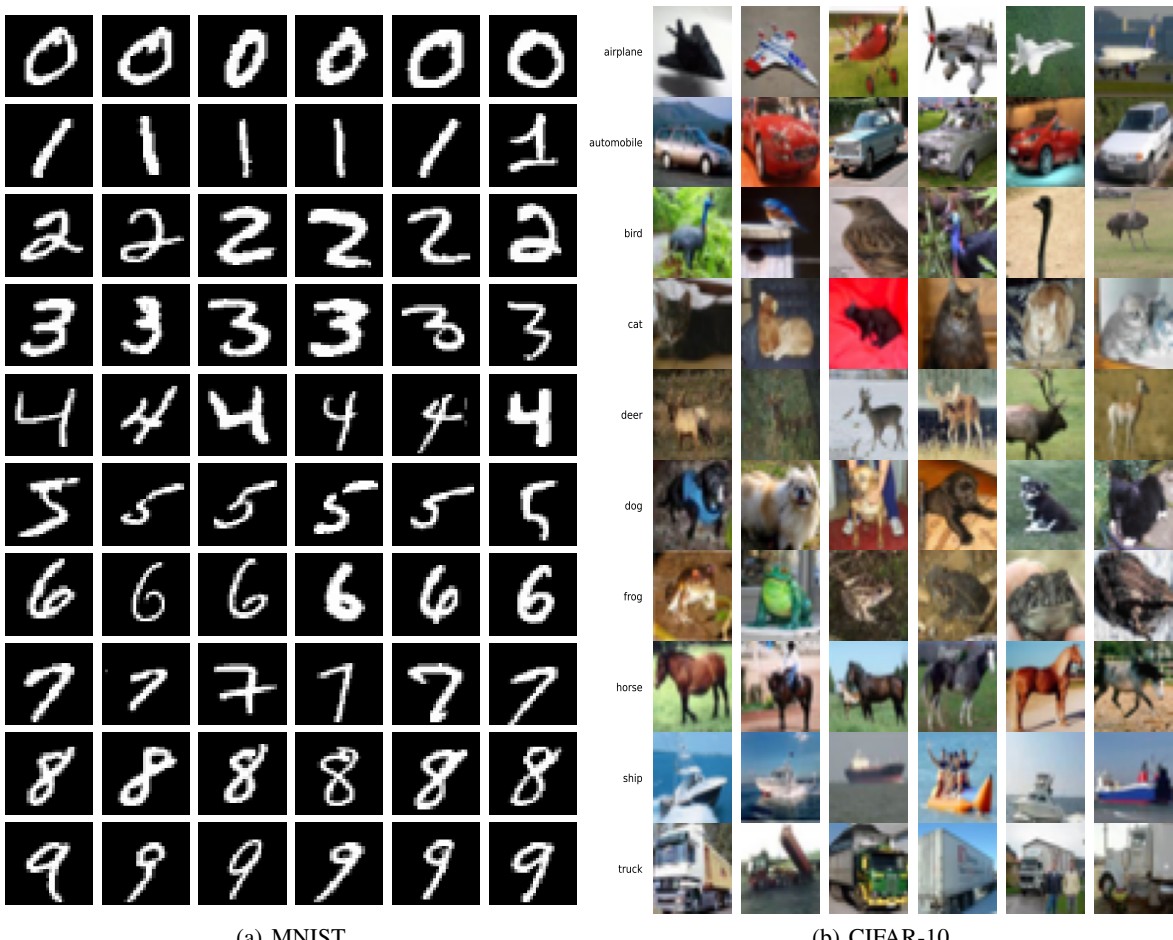

(a) MNIST                      (b) CIFAR-10

*Figure 11.* Generated samples from the proposed vMF-based diffusion model trained on (a) MNIST and (b) CIFAR-10 datasets. The samples effectively preserve class-specific information while maintaining high visual quality.

Inner Hypercone                                                    Outer Hypercone

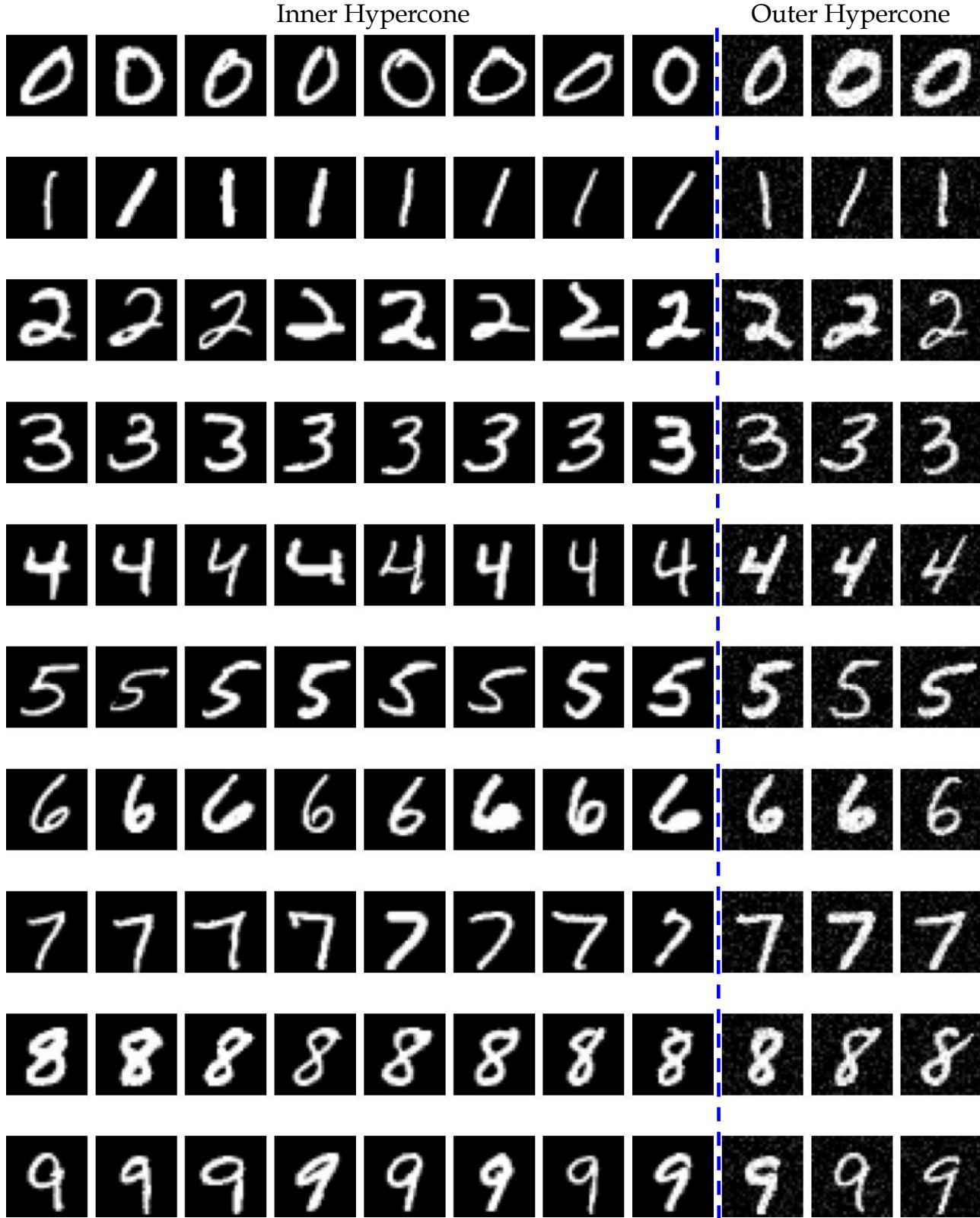

*Figure 12.* Generated samples from the inner and outer hypercones using the proposed vMF-based diffusion model trained on the MNIST dataset. Samples from the outer hypercone exhibit noticeable noise and distortions.

Inner Hypercone · Outer Hypercone

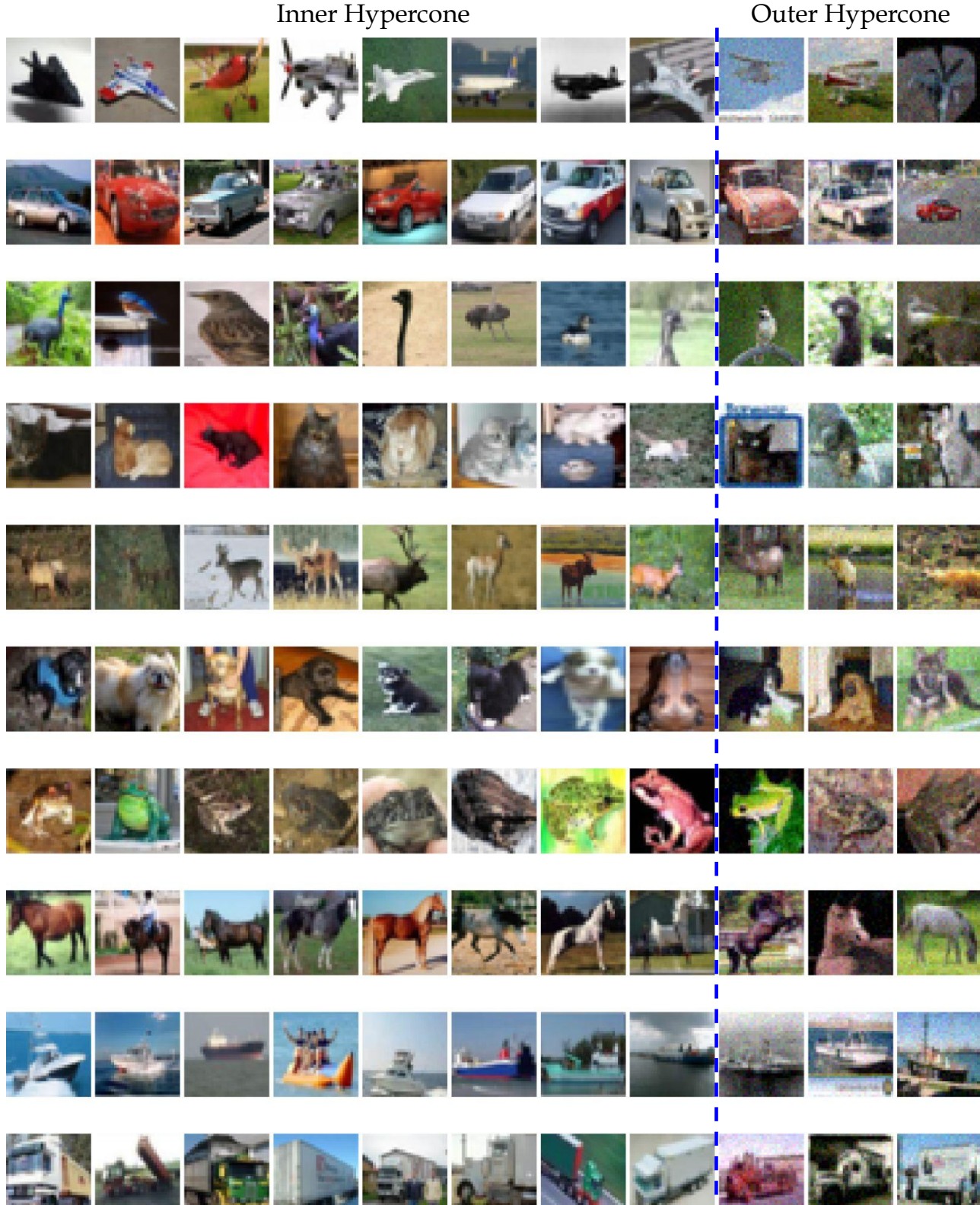

*Figure 13.* Generated samples from the inner and outer hypercones using the proposed vMF-based diffusion model trained on the CIFAR-10 dataset. Inner hypercone samples exhibit superior quality and effectively preserve class-specific information.

