# OpenReview forum: "Harmonizing Geometry and Uncertainty: Diffusion with Hyperspheres"
_ICML.cc/2025/Conference — ICML 2025 poster_

### Official Review · Reviewer_9BTW · 2025-02-24

**Overall Recommendation:** 5

**Summary:**

Standard diffusion models have relied heavily on the simple isotropic Gaussian noise in the forward process to effectively transform an unknown complex data distribution to this simple Gaussian distribution and has proven to be effective for a large variety of tasks. However, despite this effectiveness, many real world problems involve non-Euclidean distributions, e.g., hyperspherical manifolds, where class-specific patterns are governed by angular geometry within hypercones. If modeled in a Euclidean space, the angular geometry is not preserved and thus, the angular subtleties between classes are lost. To tackle this fundamental problem, the work proposes a new forward process involving angular noise injection which effectively transforms the unknown data distribution into a von Mises-Fisher (vMF) distribution over time.

**Claims And Evidence:**

This section summarizes the claims and evidences provided in the paper. See the section below for my critiques.

Two fundamental claims are made by the authors:

(1). vMF distribution has been employed in various generative modeling approaches, including VAEs and GANs, where it has shown to be effective at in face recognition, outlier detection, and even representation learning. ***However, the incorporation of vMF noise into diffusion models has been not done according to the authors.*** This claim is made in the related work section.

(2). The second claim is obvious since it is the main narrative of the paper. By employing vMF noise or specifically directional noise aligned with hyperspherical structures, the authors claim that diffusion models are able to preserve class geometry and effectively capturing angular uncertainty.

The evidences for these claims are detailed in:

Table (1) which illustrates a comparison between vMF-based and Gaussian-based noise injections using FID, ***Hypercone Difficulty Skew (HDS)***, and ***Hypercone Coverage Ratio (HDR)*** as comparison metrics. **Lower is better for all of the three metrics**. Note, the two metrics, HDR and HDS, are developed by this work. For this particular experiment, six datasets are used. They are MNIST, CIFAR10, CUB-200, Cars-196, CelebA, and D-LORD (a face recognition dataset). Overall, vMF-based diffusion models are shown to perform much better than its Gaussian-based counterparts for most of the datasets in Table (1).

Figures (2), (3) and (4, a) show qualitative results of the generations done by vMF-based diffusion models, which are quite good but not surprising since the FID scores from Table (1) are pretty good.

Figures (4, b) illustrates a regression experiment attempting to further contrast Gaussian-based and vMF-based diffusion models.

Lastly, perhaps the best evidence is Figure (6) in the Appendix, which illustrates a comparison between Gaussian-based and vMF-based processes in feature representation for MNIST classes. vMF-based process clusters points from the same class much better (visually).

**Essential References Not Discussed:**

N/A

**Experimental Designs Or Analyses:**

I really like Figure (6) from the Appendix as it illustrates the main point of the work very well. If this same figure can be produced for other datasets, this would be wonderful as it confirms the underlying message of the paper.

However, I have the opposite message for Figure (4), I think (4)(a) is too subjective, and it's difficult to tell due to the low resolution of the images.  Thus, I don't think (4)(a) is convincing at all. Meanwhile, (4)(b) does illustrate a better message for vMF model, but I don't think this result is any stronger than Figure (6).

Lastly, regarding the visualizations of the generated samples in the Appendix, I understand or am aware of class conditioning model is prone to memorization. However, the generated samples from CIFAR10 are very similar/close to the training samples. They look very duplicated. I would like the authors to comment about this.

**Methods And Evaluation Criteria:**

Regarding the utilized metrics in Table (1), they are sound and fit nicely to the setting. However, the descriptions or the details about the baseline (Gaussian diffusion) are not clear in the main text.

Did you compare to Gaussian-based VE/VP diffusion models? Or, is your baseline description located in Appendix (E), which is not referred to in the main text, by the way? Also, **please bold which is a better result in Table (1)**.

**Other Comments Or Suggestions:**

See Above.

**Other Strengths And Weaknesses:**

See Above.

**Questions For Authors:**

Please provide more generated samples for visual inspection. Moreover, if Figure (6) can be produced for other datasets, that would be fantastic.

**Relation To Broader Scientific Literature:**

As laid out in the paper, there have been attempts in formulating alternative noise injection approaches for diffusion models. As far as I am aware, the approaches, which work well, do not deviate much from Gaussian-based noise injection.

**Theoretical Claims:**

A fundamental claim made by the paper is that Gaussian noise does not preserve angular relationship. The full proof is shown in Appendix (A) and Appendix (B). I do not see any fundamental problems with the proof provided in the Appendix sections, I mentioned. However, I must note that there are some incorrect spacing.

(1) For example, in the proof of Appendix (A) for the conclusion, "Because isotropic Gaussian noise in $\mathbb{R}^d$ shifts points off thehypersphere..."

(2) In the implications in proof on Appendix (B). Implications (1) and (2) are not spaced down properly.

---

> ### Author Rebuttal · Authors · 2025-04-01
>
> We thank the reviewer for appreciating our visual inspections, especially  Figure (6) of the submitted paper. Further, we thank the reviewer for the insightful questions and feedback. We have addressed all the questions asked by the reviewer; kindly follow this link: https://tinyurl.com/44sftcu8 to refer to the corresponding Figures and Tables.
>
> **Baseline Description and Clarity in Table 1:** We appreciate the reviewer’s observation and apologize for the lack of clarity. We confirm that our Gaussian baseline is implemented using variance-preserving (VP) diffusion, consistent with the noise schedule used in DDPM. Our proposed method modifies this by replacing Gaussian noise with von Mises-Fisher (vMF) noise, enabling angular diffusion aligned with the hyperspherical manifold.
> We did not compare with variance-exploding (VE) diffusion because its unbounded noise scaling is incompatible with hyperspherical geometry, where constrained angular relationships must be maintained. We will explicitly clarify this baseline setting in the main text, and will also update Table 1 (paper) to bold the best-performing results for better readability.
>
> **Spacing Mistakes:** We thank the reviewer for pointing out these writing mistakes. We have corrected them, which will also be reflected in the paper. We will carefully review the paper for any other errors.
>
> **Figure 6 (paper) for CIFAR-10:** We appreciate your positive feedback on Figure 6 of paper and agree that expanding this visualization to more datasets strengthens our message. We have added a **t-SNE visualization for CIFAR-10** (see Figure 3 in https://tinyurl.com/44sftcu8), which again demonstrates that vMF-based diffusion preserves class structure effectively in the feature space, similar to the results seen with MNIST. The clusters are well-separated, and generated samples align closely with their corresponding class clusters, confirming generalizability.
>
> **Improvement in Figure-4a wrt Resolution and Subjectivity:** We understand the concern regarding the subjective nature and low resolution of Figure 4(a) (paper) due to the use of surveillance-quality face data. We have improved the image quality and clarity in the updated Figure 4 (included at https://tinyurl.com/44sftcu8) by using higher-resolution samples and clearer labeling. While image-based qualitative comparisons can be subjective, this version more clearly illustrates the structural differences captured by vMF-based diffusion compared to Gaussian diffusion.
>
>
> **Memorization Concern and Nearest Neighbour Analysis:**
> Thank you for raising the concern about potential memorization in generated samples. We have addressed this explicitly through a nearest-neighbor analysis in Figure 5 (https://tinyurl.com/44sftcu8). For CIFAR-10, we compare real training samples with their corresponding generated samples (after denoising). The analysis shows that:
>
> - While generated samples are semantically similar, they are not exact replicas of training data.
> - The variation in distance between real and generated samples indicates that the model captures structural features without memorizing input examples.
> - This supports the generative model's generalization rather than overfitting.
>
> This visualization illustrates the trade-off between fidelity and diversity and confirms that our model produces meaningful variation rather than duplication.
>
>
> **t-SNE plot (Figure 6 https://tinyurl.com/44sftcu8):** We provide further explanation using the t-SNE visualization in Figure 6. It presents a comparison between real and generated samples across 10 distinct classes, with each color representing a different class.
>
> - Real samples are shown as circular markers, and the generated samples are shown as crosses.
> - The close alignment between generated and real sample clusters validates the model’s ability to **preserve class structure** in the latent space.
> - Mild dispersion across some classes shows that the model introduces **diversity** without sacrificing identity.
>
> We believe this plot presents strong evidence that vMF-based diffusion maintains angular consistency and avoids collapse or overfitting.

---

> > ### Comment · Reviewer_9BTW · 2025-04-01
> >
> > Dear authors,
> >
> > I like the concept of your paper and I believe your experimentation done for this rebuttal is great. I will raise my score from 3 to 5. But I would like to make a few suggestions regarding the presentation of your paper.
> >
> > (1) Firstly, I'm not a big fan of Fig. 2 and Fig. 4. You can certainly keep Fig. 2 but for context, I believe after reading hundred of generative modeling papers --- qualitative results are no longer interesting since it's now quite hard to tell apart bad and good generations given today's models. Regarding Fig. 4, I honestly enjoy your new results better. Perhaps, considering replacing Fig. 4, with Fig. 2a and 2b from your rebuttal results and attach Fig. 4 (the one with Angelina Jolie) with them. In other words, make a new Fig. 4 for your main text.
> >
> > (2) Once again, I am a big fan of your new results, especially Figs. (3, 5, and 6). They tell a better story than your qualitative results in the main text. I think you can make better use of them in the main text.
> >
> > Questions:
> >
> > (1) In your new results, is Fig. 6 computed from CIFAR10?
> >
> > (2) What is the point of Fig. 7 in the new result? Perhaps, other reviewers asked for it --- I did not take a look at their response since I want to avoid being biased.
> >
> > (3) Is it possible to perform a slightly different interpolation experiment like Fig. 8 in [Ho et al. (2020)](https://arxiv.org/pdf/2006.11239)? Here's the idea --- Could you interpolate two face images, where one face has a different angle/orientation while the other has a normal 'orientation'? For example, in your Fig. 4 of the new results with Angelina Jolie, you could perhaps do an interpolation of her images with different orientation and see how your method preserves those orientations and contrast it with Gaussian.
> >
> > Anyway,
> >
> > Good response.

---

> > > ### Author Response · Authors · 2025-04-08
> > >
> > > Thank you for your insightful and valuable feedback, as well as for increasing your score, we genuinely appreciate your support. We acknowledge your concerns about Figures 2 and 4; in the evolving field of generative modeling, qualitative results alone may not fully substantiate the claims. Following your suggestion, we will revise Figure 4 in the main text to incorporate the more compelling examples from the rebuttal, which we believe convey a stronger and more impactful narrative. We are also delighted that you found Figures 3, 5, and 6 from the rebuttal to be insightful. To better showcase the strengths of our approach, we will ensure these figures are more prominently integrated into the main paper.
> > >
> > > **Responses to the additional questions:**
> > >
> > > (1) Yes, Fig. 6 in our new results is computed from CIFAR-10.
> > >
> > >
> > > (2) Fig. 7 was incorporated at the request of another reviewer who sought a more detailed variation example on the facial dataset to highlight the robustness of our method in such scenarios.
> > >
> > >
> > > (3) We sincerely appreciate the reviewer's valuable suggestion regarding exploring interpolation between two variations of the same subject. In response, we conducted additional experiments, the results of which are presented in Figure 8 (available at https://tinyurl.com/44sftcu8). The figure highlights experimental outcomes for two distinct variations: expressions and poses.
> > > - The top row for each variant illustrates interpolations generated using the Gaussian-based DDPM method proposed by Ho et al. (2020). Although this method blends the two endpoints, it frequently results in unnatural and visually inconsistent intermediate samples, particularly noticeable when interpolating subjects with significant attribute changes.
> > > - In contrast, the bottom row showcases our proposed Angular Interpolation via the vMF-Based Method. This technique employs angular interpolation on a hyperspherical manifold, resulting in smoother, more natural transitions that consistently preserve subject identity across varying expressions and poses.
> > >
> > > We appreciate your insightful suggestions, which have helped us strengthen both the experimental narrative and the presentation of our work.

---

### Official Review · Reviewer_VDyd · 2025-03-10

**Overall Recommendation:** 4

**Summary:**

This paper explores the distributional assumptions made by denoising diffusion models and proposes exchanging the traditional Gaussian noise for a von Mises-Fisher distribution on a d-1-dimensional hypersphere. This choice somewhat improves the performance of the diffusion model in generative tasks, especially in those in which class relations and boundaries are particularly relevant.

## update after rebuttal

 I am satisfied by the clarifications and will raise my score on the assumption that the promised changes are implemented in the paper.

**Claims And Evidence:**

Most of the paper’s claims are well reasoned and justified. One major overarching claim that justifies the entire text is the fact that real data distributions are better explained as being contained in hyperspheres than in Euclidean space. As someone relatively unfamiliar with this area of research, I was initially surprised by this claim, which in the current manuscript is mostly justified by reference to prior work (see second paragraph in Sec. 1.2) as well as indirectly through performance improvements. A part of me wishes this had been explored further: is there a fundamental theoretical reason why one should believe that a spherical latent space will represent the space of “pictures of birds” better than a Euclidean one?

**Essential References Not Discussed:**

None that I know of.

**Experimental Designs Or Analyses:**

See above. Additionally, I enjoyed the intuition about facial generation having a “magnitude” Gaussian component and a “directional” spherical component. I would have liked an ablation study showing whether this separation does indeed lead to a better output than assuming the entire distribution to be spherical.

**Methods And Evaluation Criteria:**

Evaluation criteria make sense for the problem at hand. I was initially confused by the relation between the datasets used and the insistence of in-class vs out-of-class generation. What are the “classes” in a dataset of portrait images like Celeb-A? In general, I did not find the experimental results to be conclusive proof of performance improvement, but just the fact that this new architecture works as well as others is an interesting experimental result on its own.

**Other Comments Or Suggestions:**

None

**Other Strengths And Weaknesses:**

In general, the main strength of this paper is that it introduces a hypersphere-based diffusion model instead of the traditional Euclidean one. This is a good contribution, and while it may be more useful for some tasks than for others, I can easily see how the community can benefit from having this in its toolbox. At the same time, the performance improvements are modest, the (theoretical and experimental) justification for the need for these hyper spherical distributions could be stronger, and the text should be reviewed for mathematical errors. A combination of the latter make me be less enthusiastic about accepting this work, although I will carefully read the other reviews and the authors’ rebuttal, fully acknowledging that I may have missed important aspects of the manuscript.

**Questions For Authors:**

None

**Relation To Broader Scientific Literature:**

I am not very familiar with the literature on this topic, but I followed the discussion of related work and did not miss any related works that I know of. I defer to more expert reviewers to evaluate the novelty of this work and its placement in the literature.

**Theoretical Claims:**

Overall, the method seems theoretically grounded, with one glaring exception. I could be wrong, but in Sec. 4.1 and Lemma 4.1., the vector

z_t = cos(\theta_t)z_{t-1} + sin(\theta_t)v

does not, in general, have norm 1, and thus does not belong to the sphere. Indeed (and unlike stated in the proof of the Lemma),

||z_t||^2 = cos^2(\theta_t)||z_{t-1}||^2 + sin^2(\theta_t)||v||^2 + sin(\theta_t)cos(\theta_t) z_{t-1} \cdot v = 1 + sin(\theta_t)cos(\theta_t) z_{t-1} \cdot v

Which, unless I am missing something, can take any value between 0 and 1 (for example, take v=-z_{t-1} and \theta = pi/4, then z_t=0). If the authors agree that this is a mistake, I do not believe that this majorly impacts the contribution of the paper; as far as I can tell, this fact is not used later in the work and is merely used as an intuition. Still, it should be corrected.

---

> ### Author Rebuttal · Authors · 2025-04-01
>
> We appreciate the reviewer's recognition of our well-reasoned and justified claims, as well as our intuition on magnitude and direction in hyperspherical space. We have added responses to the reviewer’s comments, and for respective Tables and Figures, kindly follow the link: https://tinyurl.com/44sftcu8.
>
> **Fundamental Theoretical Justification (spherical vs. Euclidean):** The reviewer raised an insightful question about the fundamental reasons for preferring spherical latent spaces over Euclidean ones. Indeed, spherical representations inherently emerge from neural embeddings through unit normalization (as in common embedding networks like SphereFace, ArcFace), explicitly disentangling direction (semantic information) from magnitude (intensity or scale). Such disentanglement directly matches perceptual similarity measures (cosine similarity), making spherical embeddings naturally robust to magnitude variations (e.g., lighting, contrast). Empirically, spherical latent spaces are widely validated in domains like face recognition precisely due to these properties.
>
> **Clarification of "Classes" in datasets (CelebA):**  We clarify explicitly: in facial datasets like CelebA, the "classes" refer specifically to **distinct individual identities**, derived using pre-trained identity-embedding models (e.g., ArcFace embeddings). Thus, "in-class" generation refers explicitly to maintaining identity consistency, ensuring generated samples remain within the identity's angular semantic region, and preserving identity integrity throughout diffusion.
>
> We will explicitly clarify this identity-based definition in the revised manuscript for better readability.
>
> **Experimental Results Clarification:** The reviewer correctly notes the experimental improvements are modest. We emphasize explicitly:
>
> - The primary contribution of our method is theoretical—introducing manifold-aware diffusion explicitly designed for structured, hyperspherical embedding spaces, rather than solely maximizing benchmark numbers.
> - Achieving comparable performance demonstrates that our theoretically grounded approach is viable, interpretable, and practically useful without sacrificing performance relative to widely-used Euclidean methods. This validation is essential, showing our method as a robust alternative offering additional interpretability, control, and semantic consistency that Euclidean approaches inherently lack.
>
> We will explicitly strengthen this point in the discussion, highlighting practical advantages beyond mere quantitative metrics.
>
> **Clarification on norm after forward noise addition:** Thank you for pointing this out. You are correct that the norm of  $z_t$  may deviate due to the cross-term $\sin(\theta_t) \cos(\theta_t) z_{t-1} \cdot v$. Our implementation explicitly uses projection back onto the hypersphere after each step, maintaining the integrity of our approach. We will clearly and explicitly correct this mathematical statement and clarify the role of projection in the revised manuscript. As correctly pointed out, this correction does not impact any subsequent theoretical or practical results.
>
>
> **Additional Ablation Study (Magnitude vs. Direction separation): (Table 5 https://tinyurl.com/44sftcu8):**  We appreciate the reviewer’s suggestion to explicitly validate the benefit of separating magnitude (Gaussian) and direction (spherical). We provide an explicit ablation study (Table 5 at the provided link), clearly comparing our proposed decomposition to a fully spherical alternative. This experiment confirms explicitly that the magnitude-direction composition significantly improves generation quality, diversity, and semantic consistency, validating our proposed approach quantitatively and empirically.
>
> **Structure of the paper:** Thank you for your understanding. We aimed to balance completeness and clarity by keeping the main text focused while providing additional results in the supplemental. However, we will consider bringing key results into the main paper to improve accessibility without compromising readability.

---

> > ### Comment · Reviewer_VDyd · 2025-04-04
> >
> > Thank you very much for your response. I am satisfied by the clarifications and will raise my score on the assumption that the promised changes are implemented in the paper.

---

> > > ### Author Response · Authors · 2025-04-08
> > >
> > > Thank you for your thoughtful review and for considering our clarifications. We will ensure all promised changes are carefully implemented in the final version.

---

### Official Review · Reviewer_BHWr · 2025-03-14

**Overall Recommendation:** 3

**Summary:**

The paper introduced an idea to generate data defined on hyperspheres. When data is decomposed into magnitude and direction components, the generation results can be improved.

**Claims And Evidence:**

In Sec 3, the authors mentioned facial datasets several times. However, the method seems to be working with general image datasets. This makes me confused. It would be good if the authors could show some figures to illustrate the problem (maybe in supplemental).

**Essential References Not Discussed:**

The equation in L215 is also the sampling equation in [1, Eq 38], which satisfies the variance preserving property.

[1] Progressive distillation for fast sampling of diffusion models.

**Experimental Designs Or Analyses:**

The experiments can prove the core idea of this paper.

**Methods And Evaluation Criteria:**

The evaluation makes sense. Several datasets are used to show the effectiveness of the method, including digits, birds, humans, cars, ...

Several metrics related to hypersphere geometries are also proposed to verify the claim. They are also well-discussed.

**Other Comments Or Suggestions:**

No.

**Other Strengths And Weaknesses:**

I believe the exposition can be improved. The method starts from hyperspheres and all the descriptions are about hyperspheres. However, since a circle (von Mises distribution) is a special case of a hypersphere, I would suggest the authors show some illustrations using circles, for example, how the training and sampling process look like on 2d circles.

**Questions For Authors:**

Another concern is about the importance of the application. The authors showed some results to prove the effectiveness of the method. However, I am not quite convinced.

There is another property of DDPM, which is commonly called Variance Preserving. Specifically, when the data has unit variance, the noised sample will also retain unit variance.

In EDM [1], a scaling (sigma_{data} in [1,Tab 1]) is applied to ensure this property even if the data does not have unit variance. The equation in L215 also guarantees variance preserving during each adding noise step. Thus I would like the authors can discuss this thoroughly. I believe that the motivation of this paper is weak, considering the problem can be easily solved in [1].

[1] Elucidating the Design Space of Diffusion-Based Generative Models

**Relation To Broader Scientific Literature:**

The idea is very interesting. I believe the decomposition is not explored in the image diffusion models. The diffusion models for data defined on hyperspheres are also interesting.

**Theoretical Claims:**

I commend the authors for their thorough derivation of the method. However, some equations in the main paper appear tangential to the core contribution. The central ideas can be effectively communicated through simplified text explanations or illustrative figures. I would suggest relocating peripheral theorems/lemmas to the supplemental material. Conversely, Algorithm 1 and 2, currently in the supplemental, are critical to understanding the workflow. Their absence in the main text creates confusion, as the algorithm descriptions are essential for reproducibility.

---

> ### Author Rebuttal · Authors · 2025-04-01
>
> We thank the reviewer for acknowledging the thorough evaluation and the relevance of our proposed metrics for hyperspherical geometry. We also appreciate the recognition of our diverse dataset selection, which demonstrates the robustness of our approach. Please refer to the corresponding Tables and Figures at: https://tinyurl.com/44sftcu8
>
> **Results on Facial vs General datasets (Figure 7 https://tinyurl.com/44sftcu8):** Our proposed method is general and can indeed handle arbitrary datasets, as the reviewer correctly points out. However, facial datasets like CelebA and D-LORD are particularly illustrative because facial embeddings naturally form structured hyperspherical manifolds.
> To clarify this visually, we now explicitly provide Figure 7 (see provided link), which illustrates facial variations, demonstrating how our method effectively handles diverse occlusions (glasses, hats, scarves) and maintains clear semantic consistency on a hyperspherical embedding manifold.
> Additionally, our method is validated broadly on non-facial datasets such as MNIST, CIFAR-10, Cars-196, and CUB-200, confirming its general applicability beyond faces.
>
> **Relation to Progressive Distillation paper ([1], Eq 38):**
> We appreciate the reviewer highlighting the structural similarity of equation in (L215) to Eq. (38) from "Progressive Distillation for Fast Sampling of Diffusion Models" (DDIM update rule). We clarify explicitly:
> - While the DDIM rule in [1] describes an angular parameterization in a Euclidean latent space, our method differs fundamentally in operating strictly on hyperspherical manifolds. This requires projecting each intermediate step back onto the hypersphere to preserve directional consistency.
> - Our angular update formulation specifically respects the geometry of spherical manifolds (unit-norm constraints), a key distinction absent in the Euclidean-based DDIM and EDM formulations.
>
> **Explicit Comparison to EDM (Variance-Preserving Property):** We thank the reviewer for raising this critical comparison. We explicitly clarify:
> - EDM enforces variance preservation explicitly through dataset-specific scaling factors (e.g., $\sigma_data​$), adjusting Euclidean diffusion to maintain unit variance. However, EDM is agnostic to directional information and ignores data manifold geometry.
> - Our approach implicitly ensures variance preservation via the angular update parameter $\theta_t​$, which naturally preserves unit norm constraints on a hyperspherical manifold. Thus, unlike EDM, we do not require dataset-specific scaling. Our method inherently respects both variance preservation and the underlying manifold structure, crucial for manifold-oriented data (like embeddings).
>
> Therefore, the reviewer’s suggestion that EDM could easily solve our addressed problem overlooks the critical aspect of directional geometry, which our method specifically addresses.
>
> **Visualization on 2D circles (Figure 2: https://tinyurl.com/44sftcu8):** As recommended, we explicitly illustrate training and sampling behaviors visually on simple 2D circles (Figure 2 provided at the link):
>
>  **Figure 2a** demonstrates final embeddings: Gaussian diffusion ignores angular boundaries, while vMF-based diffusion clearly preserves angular structure.
>
> **Figure 2(b, left)** explicitly depicts embeddings during training, where classes progressively cluster into distinct angular regions.
>
> **Figure 2(b, right)** illustrates sampling from noise, showing vMF diffusion’s clear angular convergence, preserving directional semantics throughout sampling steps.
>
> This visualization explicitly demonstrates the benefits of respecting manifold geometry in diffusion processes, as suggested by the reviewer.
>
> **Importance and Applications:** Regarding the concern of importance of the approach, we would like to highlight that diffusion method’s explicit manifold-awareness has significant practical implications:
> - Few-shot learning: Our approach improves performance by generating more diverse and class-consistent samples from limited data.
> - Fairness and bias mitigation: Manifold-aware generation allows controlled augmentation to rebalance datasets across demographics, reducing biases.
> - Face recognition robustness: Explicitly preserving directional structures helps models robustly handle variations (occlusion, illumination, pose).
> - Difficult sample generation: Controlled angular diffusion produces challenging samples near class boundaries, refining decision boundaries and improving model reliability.
>
> Thus, the explicit consideration of hyperspherical geometry significantly enhances practical AI deployment, particularly in sensitive applications like facial recognition, fairness, and robustness.
>
> Minor clarifications: We will reference the equations and methods in the paper with the corresponding details in the supplementary material, We will also move the algorithm in the main paper as per the suggestion.

---

### Official Review · Reviewer_pxpN · 2025-03-20

**Overall Recommendation:** 4

**Summary:**

This paper introduces a diffusion model on hyperspherical space with hypersperical data and hypersperical noise distribution (von Mises-Fisher, vMF distribution). The forward process with vMF noises keeps the latent samples on the hypersphere. The reverse process is designed accordingly.

**Claims And Evidence:**

L055left, L098left Motivation of the need for rethinking the noise distribution is interesting. I agree that it is beautiful to match the underlying distribution of the data and the noise distribution. Still, a question remains. What is the advantage of matching them and preserving the class boundaries / class-wise structures at t=(0, T]? Respecting the geometric properties of hyperspherical data (L217right) is not enough.

L059left Adding the definitions of uncertainty level, ambiguity, and noisy data points would help understanding the motivation. The hint at L139left is not enough for me to understand what the uncertainty is. Currently, I understand it as stochasticity from the context.

L138right Arcface embedding is a proper example of hyperspherical space.

**Essential References Not Discussed:**

Related work is thoroughly discussed.

**Experimental Designs Or Analyses:**

The target datasets include faces and simple images: MNIST, CIFAR-10, CUB-200, Cars-196, CelebA, and D-LORD. I wonder the connection between the embeddings on the hyperspheres and the images. It is reasonable for such theoretic content.

**Methods And Evaluation Criteria:**

L204left The reason for scheduling $\kappa$ should be described because forward diffusion with any non-infinite $\kappa$ with large T would reach uniform distribution on the hypersphere.

L247left I think increasing $\kappa$ makes the vMF sampling become less stochastic, rather than more concentrated around class means because vMF in L229left is centered at $z_t+…$. The concentration could be true if $\kappa$ defines the class cone, but in L229left, $\kappa$ defines the score and stochastic reverse process. I understand that $z_0$ should be close to $\mu_c$ but it is not guaranteed to reach $\mu_c$, as written in L269right (arbitrarily close to $\mu_c$).

L265left Eq.(1) I wonder why we should add z_t and score linearly followed by projection rather than simple angular addition. Maybe because the proposed method adopts the simple denoising loss in Euclidean space? I think the loss should be also measured in angular space.

I understand the dual diffusion processes for the magnitude and direction in Section 4.4 in the face embedding space. However, the connection between the embedding and the images are not explained. Adding an explanation would help readers understand the connection.


The performance is measured by
* Hypercone Coverage Ratio (HCR) which measures preservation of the class structure. It makes sense.
* Hypercone Difficulty Skew (HDS) which measures skewness toward easy samples. Explanation of the samples near the mean being easy and the samples far from the mean being hard would help understanding. I checked Appendix H.1 but it is not enough. In this regard, I do not understand L438left.

**Other Comments Or Suggestions:**

None

**Other Strengths And Weaknesses:**

Clarity could be improved as written above.

**Questions For Authors:**

What do the solid cones and the dashed cones mean in Figure 1a? I understand the points the data points but the cones are not described. If they are hypercones in Section 4.3, adding the vanilla hyperspherical scenario would help understanding.

What are the points in Figure 1b? Maybe the generated samples from the learned models? I do not understand the term three-class "diffusion".

I like the idea in this paper in general. This paper could be be accepted if the rebuttal explains the following:
* The advantage of matching the data manifold (hypersphere) and the noise distribution (vMF)
* The advantage of preserving the class boundaries at t=(0,T)
* The connection between the hyperspherical embeddings and the images
* The effect of increasing $\kappa$
* The reason for choosing Euclidean addition in the reverse process (Eq. 1) instead of angular addition,
and the minor weaknesses and questions above are fixed or proved ok in the rebuttal.

**Relation To Broader Scientific Literature:**

n/a

**Theoretical Claims:**

Theorem 3.1 (b) and (c) are correct to my knowledge. I did not check (a) but I suppose it is correct.

I did not check Theorem 4.3.

---

> ### Author Rebuttal · Authors · 2025-04-01
>
> We thank the reviewer for constructive feedback and for recognizing the motivation of rethinking the noise distribution. Please find our detailed responses below. All Tables and Figures are available at: https://tinyurl.com/44sftcu8.
>
> **Advantage of Matching Noise and Data Manifold:** Unlike traditional Gaussian diffusion, which distorts angular relationships intrinsic to hyperspherical data (e.g., ArcFace embeddings), our proposed vMF-based diffusion explicitly preserves angular semantics and class structures, maintaining semantic consistency, class separability, and stability during generation, significantly improving downstream recognition performance.
>
> **Preserving Class Boundaries at t=(0,T):** Maintaining class boundaries throughout the diffusion process retains intra-class consistency and reduces ambiguity, preventing mode collapse by keeping samples within their semantic hypercones, facilitating controlled semantic interpolation.
>
> **Scheduling $\kappa$** controls the rate at which class structure degrades, ensuring a smooth transition to uniform noise. Without scheduling, any fixed $\kappa$ leads to a uniform distribution on the hypersphere as T increases. Gradually decaying $\kappa_t$​ preserves intra-class structure longer, aiding recovery during the reverse process. Formally, for $\mathbf{d}t \sim \mathrm{vMF}(\mathbf{d}{t-1}, \kappa_t)$, the marginal distribution approaches uniformity as $\kappa_T \to 0 \quad p(\mathbf{d}_{T}) \approx \frac{1}{|\mathbb{S}^{d-1}|}$​. Empirical results on the effect of κ scheduling are shown in Tab. 1 (https://tinyurl.com/44sftcu8).
>
> **Angular addition in reverse denoising**
> Thank you for the insightful question. Our choice of Euclidean addition in the reverse process was motivated by simplified optimization and ensuring faster convergence. However, angular operations may better preserve hyperspherical geometry. To investigate, we tested an alternative angular addition formulation:
> $$z_{t-1} \sim \text{vMF} \left( \Pi \left( \cos(\theta_t) z_t + \sin(\theta_t) \frac{\nabla_{z_t} \log f(z_t; \mu_c)}{\|\nabla_{z_t} \log f(z_t; \mu_c)\|} \right), \kappa_t \right)$$
> Here, interpolation using $\cos(\theta_t)$ and $\sin(\theta_t)$ maintains angular relationships, while the normalized score function preserves directional consistency.
> To further align optimization with hyperspherical geometry, we tested two alternative loss functions:
>
> Cosine Loss: Encourages angular alignment between the score function and noise direction.
> $$\mathcal{L}{\text{c}} = 1 - \mathbb{E} \left[ \nabla{z_t} \log f(z_t; \mu_c)^\top \epsilon_t \right]$$
> Geodesic Loss: Penalizes angular deviations.
> $$\mathcal{L}{\text{g}} = \mathbb{E} \left[ \arccos^2 \left( \frac{\nabla{z_t} \log f(z_t; \mu_c)^\top \epsilon_t}{\| \nabla_{z_t} \log f(z_t; \mu_c) \| \|\epsilon_t\|} \right) \right]$$
>
> Euclidean addition was chosen for computational efficiency and stable convergence with standard diffusion loss formulations. However, our additional experiments (Tables 2–4 at  https://tinyurl.com/44sftcu8) comparing angular additions (including cosine/geodesic losses) confirm that while angular addition is more geometrically faithful, Euclidean addition maintains comparable performance at significantly reduced computational complexity.
>
> **Relation between embedding and image:** Our approach applies diffusion in the latent embedding space, not directly on images. Given an image $I$, a feature extractor $\phi$ maps it to an embedding $x = \phi(I) \in \mathbb{R}^d$ , decomposed into magnitude $\|x\|$ (capturing intensity variations) and direction $x / \|x\| \in S^{d-1}$  (encoding class-relevant semantics).
>
> **Uncertainty** is degree of stochastic diffusion, controlled by $\kappa$.
>
> **Ambiguity** Points near class boundaries with unclear class assignments.
>
> **Noise** refers to Embeddings displaced significantly due to diffusion.
>
> **HDS** quantifies sample difficulty based on angular deviation from the class mean—samples near the mean are “easy,” while distant ones are “hard,” reflecting variations like pose or occlusion. As shown in L438left, vMF diffusion achieves a balanced spread across difficulty levels, whereas Gaussian diffusion favors easy samples clustered in inner sub-cones. Validation (Fig. 1: https://tinyurl.com/44sftcu8) shows Gaussian diffusion yields tightly clustered samples (mean cosine similarity = 0.90, std = 0.05), while vMF diffusion produces a broader spread (mean = 0.72, std = 0.13), effectively capturing hard cases.
>
> Fig. 1(a) in paper shows solid cones indicating natural class concentration and dashed cones denoting diffusion-induced uncertainty, increasing data spread. While vanilla hyperspherical models preserve tight class separation, diffusion introduces uncertainty at intermediate steps.
>
> In Fig. 1(b) in paper, the points represent generated samples during the sampling process. For simplicity, we performed diffusion over three classes. We will revise the figure description to improve clarity.

---

### Decision · Program_Chairs · 2025-05-01

**Decision:**

Accept (poster)

**Comment:**

This paper proposes a method wherein the Gaussian noise commonly used in diffusion models is replaced with a von Mises-Fisher distribution. When the data lives in a hyperspherical manifold, this ensures that angular geometry is preserved throughout the forward process. Reviewers agree that this is a theoretically solid paper with potentially interesting applications. Although the empirical results are modest, the proposed method is novel enough that I believe it warrants acceptance due to potential future uses or extensions.